# Construction and iterative redesign of *synXVI* a 903 kb synthetic *Saccharomyces cerevisiae* chromosome

Hugh D. Goold[1,2,13] ✉, Heinrich Kroukamp[2,12,13] ✉, Paige E. Erpf[2], Yu Zhao[3], Philip Kelso[2], Julie Calame[2], John J. B. Timmins[2], Elizabeth L. I. Wightman[2,12], Kai Peng[2], Alexander C. Carpenter[2,4], Briardo Llorente[2,5], Carmen Hawthorne[2], Samuel Clay[2], Niël van Wyk[2,6], Elizabeth L. Daniel[2], Fergus Harrison[2], Felix Meier[2], Robert D. Willows[2], Yizhi Cai[7], Roy S. K. Walker[2], Xin Xu[2], Monica I. Espinosa[2], Giovanni Stracquadanio[8], Joel S. Bader[9], Leslie A. Mitchell[3], Jef D. Boeke[3,10,11], Thomas C. Williams[2,4], Ian T. Paulsen[2,5] ✉ & Isak S. Pretorius[2] ✉

The Sc2.0 global consortium to design and construct a synthetic genome based on the *Saccharomyces cerevisiae* genome commenced in 2006, comprising 16 synthetic chromosomes and a new-to-nature tRNA neochromosome. In this paper we describe assembly and debugging of the 902,994-bp synthetic *Saccharomyces cerevisiae* chromosome *synXVI* of the Sc2.0 project. Application of the CRISPR D-BUGS protocol identified defective loci, which were modified to improve sporulation and recover wild-type like growth when grown on glycerol as a sole carbon source when grown at 37 °C. LoxPsym sites inserted downstream of dubious open reading frames impacted the 5' UTR of genes required for optimal growth and were identified as a systematic cause of defective growth. Based on lessons learned from analysis of Sc2.0 defects and *synXVI*, an *in-silico* redesign of the *synXVI* chromosome was performed, which can be used as a blueprint for future synthetic yeast genome designs. The *in-silico* redesign of *synXVI* includes reduced PCR tag frequency, modified chunk and megachunk termini, and adjustments to allocation of loxPsym sites and TAA stop codons to dubious ORFs. This redesign provides a roadmap into applications of Sc2.0 strategies in non-yeast organisms.

Design and construction of synthetic genomes, enabled by decreases in DNA synthesis costs, has allowed scientists to develop new perspectives on and opportunities for biological design[1] whilst offering a completely new means of interrogating biological systems and the genetics that underlie them[2]. Synthetic genomes can offer potential to enhance the biomanufacturing of important biological components such as proteins and metabolites[3]. Inclusion of symmetrical lox sites (*loxPsym*) adapted from the Cre-Lox recombination system in

intergenic loci allows for stochastic chromosomal rearrangements of genomes on expression of Cre recombination, referred to as SCRaMbLE (Synthetic Chromosome Rearrangement and Modification by LoxP-mediated Evolution)[4]. The design of SCRaMbLE has allowed for randomization of genomes, and SCRaMbLE-ready genetic elements, such as neochromosomes or entire metabolic pathways[4,5].

Use of microbes in biomanufacturing has underpinned the sustainable use of production of enzymes, and metabolites outside of

A full list of affiliations appears at the end of the paper. ✉e-mail: hugh.goold@dpi.nsw.gov.au; heinrich.kroukamp@microbiogen.com; ian.paulsen@mq.edu.au; sakkie.pretorius@mq.edu.au

**Fig. 1 | Schematic construction and debugging of *synXVI*.** Five haploid strains, 5, 6, 8, 10 and 11 were constructed on BY4741 backgrounds (coloured in blue) with contiguous synthetic DNA (coloured in red). Switching mating types[20] facilitated meiotic crossover exploitation to generate chromosome arms. Strain 9, bearing all *synXVI* synthetic DNA for the left chromosome arm to megachunk O, was generated by crossing Strains 5 and 6 to yield Strain 7, which was then crossed with Strain 8. Strain 12, bearing all synthetic DNA from *synXVI* right chromosome arm plus megachunks O to Q was generated by crossing Strain 10 and Strain 11. Strain 15 bearing synthetic DNA from chunk B3 to BB4 was generated by crossing Strain 12 and Strain 9. Debugging, discrepancies, and reinsertion of missing synthetic chunks and megachunks were completed in Strain 15. Subsequently the missing subtelomeric regions on *synXVI* L-arm were reinserted, resulting in fully synthetic Strain 30. The centromere *CEN16* is marked in magenta on chunk Q4.

The technology of synthetic genomics can help to secure supply chains in future situations where changing climates, further global pandemics, and conflict threatens the availability of critical and conventional feedstocks of food and pharmaceuticals. Application of synthetic genome design has, to date, been limited largely to bacterial and viral projects[13,14]. The Sc2.0 project represents an important step towards application of genome-scale biological design to eukaryotic systems and represents a gateway project towards design of other eukaryotic genomes. In keeping with the design principles outlined in the Sc2.0 genome synthesis project[15], the 948,066-bp chromosome XVI of *Saccharomyces cerevisiae* strain BY4741 was redesigned and constructed. Integration of 344 loxPsym sites, interspersed between non-essential genes, removal of 17 tRNA genes (totalling 1,338 bp), conversion of 127 TAG to TAA stop codons, removal of 15 introns, 15,493 bp of recoded PCR tags, 1,374 bp of recoded restriction enzyme target sites, and 10,047 bp of deleted repeats has resulted in a synthetic chromosome XVI (*synXVI*) of the Sc2.0 genome comprising of 902,994 bp[15]. The synthetic sequence was synthesized on 116 pUC vectors, which were integrated into six separate yeast strains, comprising a total of 28 physical contigs which were consolidated into a final *synXVI* strain by successive rounds of meiosis following previously detailed approaches[16].

Sc2.0 is the world's first project of design and synthesis of a eukaryotic genome. Projects spawned from Sc2.0 demonstrate many of the benefits of a designer eukaryote; Hi-C analyses of strains that underwent SCRaMbLE were insightful in characterising how chromosomes are packaged in the nucleus[17,18]. SCRaMbLE has demonstrated a high level of utility in degrading problematic wastes and guiding reverse engineering[12]. Individually this work, and others show the utility of hosting tRNAs in arrays external to chromosomal DNA to allow for larger scale and randomized approaches to chromosomal modifications[19,20]. Karyotype engineering, by changing eukaryotic genomes either to circular chromosomes such as with the perfect designer ring chromosome V[21], to the $n = 2$ yeast genomes[22,23] both demonstrate the possibility of engineering yeast genomes in, and out of, the framework of Sc2.0, demonstrating the plasticity of the yeast genome. Genetic speciation as a means of biocontainment[24,25], genetic variant generation from a single engineering project[4], new genomes which could lead to increased tractability, and organisms capable of functioning in ways their parental strains never would have evolved are all clear examples of the utility of synthetic genomics. Moving from the Sc2.0 project towards plant cell 2.0 or mammalian cell 2.0, will require a detailed meta-analysis of what has worked and what has not worked in the model Sc2.0 project. SwAP-in may be suitable for an organism with a doubling time of 90 minutes but may not be viable for an organism such as *Ostroeoccocus tauri* which has a doubling time of 1.4 days. Cas9-independent homologous recombination based megachunk insertion might be optimal for an organism such as *S. cerevisiae*, but mammalian cells, which have low homologous recombination frequencies independent of Cas9 double strand breaks and are diploid, may require alternative approaches[26]. Given growing interest in synthetic genomics with the commencement of projects with plant genomes and other organisms, we outline lessons learnt from *synXVI* and Sc2.0 more broadly and present a refined version of Syn XVI as an example of iterated genome design parameters.

## Results

### Design, construction, and debugging of a synthetic chromosome

Construction of *synXVI* of the Sc2.0 genome was undertaken in a tandem approach (Fig. 1). Strains 5, 6, 7, 10, and 11 (bearing synthetic DNA megachunks A-G, G-J, J-O, P-W, and W-BB, respectively) were built as separate strains, sequenced, and then backcrossed. *SynXVI*-L (Strain 9 bearing synthetic DNA A to O) and *synXVI*-R (Strain 12 bearing synthetic DNA from O to BB) were combined by crossing to create Strain 15,

normal supply chains. This has been classically demonstrated using the biosynthesis of artemisinic acid in yeast[6], and in more recent times, complex medicines, such as vinblastine[7], as well as peptide-based hepatitis vaccines[8]. The ability to randomize a genome to fast-track removals of physiological bottlenecks or for rapid pathway optimization[3] can be seen in several examples where SCRaMbLE has been applied to increase titre of final product yield or robustness of strains[9–12]. SCRaMbLE provides a key point of difference to classic EMS (Ethyl-Methyl-Sulfonate) and UV (Ultra-Violet) mutagenesis by enabling the exploration of a different evolutionary landscape. EMS or UV mutagenesis typically result in SNPs or indel mutations, while SCRaMbLE enables large scale events such as deletions, duplications, inversions, and rearrangements, which predominantly happen only over evolutionary timescales. Thus, EMS or UV mutagenesis are well suited for selecting for simple phenotypes, such as increasing the expression of a single gene or tweaking the kinetics of a single enzyme. In contrast SCRaMbLE opens the possibility of evolving more complex phenotypic changes that alter expression of multiple genes.

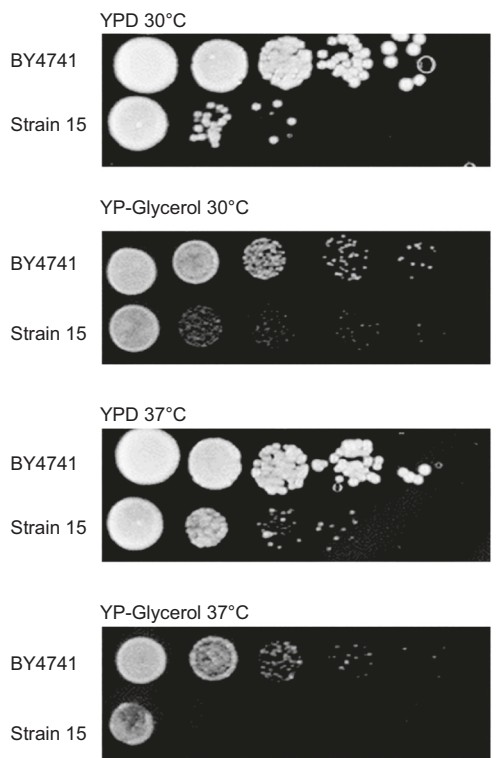

**Fig. 2 | Strain harbouring synthetic megachunk exhibits reduced growth.**
BY4741 and Strain 15 harbouring synthetic chunks B3, B4 and megachunks from C to BB4 exhibited reduced growth on non-fermentable carbon source (glycerol) and at 37 °C, when compared to a BY4741 control.

which now encoded synthetic chunks B3 to BB4. Sequencing read depth revealed multiple assembly errors including tandem chunk and megachunk duplications, deletions, pUC vector insertions, missing TAG-TAA codon swaps, and the presence of wild-type specific introns, (Supplementary Fig. 1).

### F3 generation backcross for defect identification
The strain harbouring *synXVI* (Strain 15 chunk B3 to megachunk BB) grew poorly (Fig. 2), with defects seen at 37 °C and at 30 °C under respiratory growth conditions (YP-Glycerol medium). A high frequency of rho⁻ mutant formation was observed. After selective backcrossing, whole genome sequencing was conducted on 15 colonies from the F3 progeny that exhibited poor growth on YP-Glycerol at 37 °C, and 15 colonies with wild-type growth on YP-Glycerol at 37 °C. Loci from F3 to I4 and X3-X4 had low frequencies of synthetic DNA in strains with no growth defect, and high frequencies of synthetic DNA in strains with a growth defect (Fig. 3A), indicating possible synthetic loci responsible for the aberrant growth phenotypes.

### Employing the CRISPR-D-BUGS method to fine-map synthetic defects
To isolate the synthetic features responsible for growth defects in these loci, the CRISPR D-BUGS methodology was applied[27] (Supplementary Fig. 2). Strain 15 (synthetic DNA from chunks B3 to BB4) was separately mated with two BY4742 strains with the *URA3* gene inserted proximally to the right and left telomeres to yield Strain 24 (*URA3* in the intergenic space between *YPR194C* and *YPR195C* corresponding to chunk BB2 near the right telomere) and Strain 25 (*URA3* at the intergenic locus between *YPL227C* and *YPL274W* corresponding to chunk A2 near the left telomere). The growth of the two diploid strains was compared to Strain 15 and the haploid strain BY4741, to compare for wild-type fitness (Supplementary Figs. 3 and 5). Diploid strains (Strain

27 for D-BUGS for megachunks A-N and Strain 28 for D-BUGS for chunks P to BB) were transformed with the pYZ555[27] plasmid encoding Cas9 under control of pGal1 with an sgRNA sequence targeting a wild-type locus on chromosome XVI. Each targeted locus corresponded to a recoded synthetic PCR tag for every megachunk of the *synXVI* chromosome (PAM sequence data available Supplementary data 1). Cas9 expression was induced overnight by growth in YP-Galactose, and cells with wild-type chromosome arm replacements were isolated by growth on 5-FOA. Four strains for each megachunk were then screened using dilution spot assays on YPD and YP-Glycerol at 30 °C and at 37 °C. Two loci were implicated based on the results obtained from strains with differential fitness within populations produced from the CRISPR-D-BUGS protocol (Supplementary Fig. 2). These data were consistent with previous backcross data, identifying the growth defect loci at chunks X4 and I4 (Fig. 3A and B).

### A defect associated with *CTR1*
DNA sequencing was conducted on CRISPR D-BUGS strains generated to investigate the defect localised to megachunk X. Strains with fitness comparable to parental Strain 24 in glycerol as a growth medium, and strains with impaired growth on YP-Glycerol at 30 °C revealed chromosomal crossover events adjacent to the major copper transporter *YPR124W* (*CTR1*) encoded on X4 (Supplementary Figs. 3 and 6). The fitness defect on YP-Glycerol medium suggested a challenge affecting mitochondria and thus, *CTR1* made sense as a possible candidate gene for the fitness defect.

The original design of *synXVI* erroneously recoded the entire *CTR1* CDS, the primary copper transporter of *S. cerevisiae*. Knockout mutants of *CTR1* are known to have a wide variety of phenotypes, including a respiratory defect that can be rescued by addition of copper sulfate. The pervasive recoding of the entire *CTR1* CDS was a result of an error in GeneDesign's RepeatSmasher module which was designed to recode only the tandem repeats in certain coding sequences (such as those in *CTR1*) for the purpose of reducing repetitiveness and making DNA easier to synthesize[15]. To determine whether the modified *CTR1* contributed to defective growth, the synthetic *CTR1* sequence including flanking loxPsym sites were introduced to a BY4741 strain (Strain 30), resulting in poor growth at 30 °C on glycerol that was reversed by the addition of copper sulfate to the media (Fig. 3C). To identify the exact nature of the change to the synthetic *CTR1* gene responsible for the fitness defect, a series of spot assays and transcriptional reporter assays were conducted on individual *CTR1* variants introduced into the wild-type strain (Fig. 3D). To our surprise, introduction of the pervasively recoded *CTR1* CDS and introduction of the *CTR1* 3' loxPsym site into BY4741 strains revealed no impaired growth. However, introduction of the loxPsym site assigned to *YPR123C*, an overlapping dubious ORF, revealed impaired growth at 30 °C on non-fermentable carbon sources (Fig. 3C). A series of reporter plasmids bearing different sequence configurations to evaluate what sequence variations might be responsible for the growth defect caused by potential changes in *CTR1* expression were built. Cell populations with a pRS413 plasmid expressing *GFP* with the native *CTR1* 5' UTR were compared to populations of strains with both the loxPsym sites assigned to *AXL1* and *YPR123C*, the *YPR123C* loxPsym site alone, or the *AXL1* loxPsym site, showing that the presence of both of these loxP sites altered transcription of *CTR1*; expression of GFP under the synthetic *CTR1* 5' UTR resulted in a lower median GFP intensity of the GFP+ gated cell population compared to the population when GFP was expressed with the wild-type *CTR1* 5' UTR (Fig. 3D, Supplementary Fig. 4).

### A defect associated with *GIP3*
To identify the defective locus between megachunk F and megachunk I, CRISPR D-BUGS was performed to generate strains to investigate potential defects between loci F to I. A diploid strain was identified

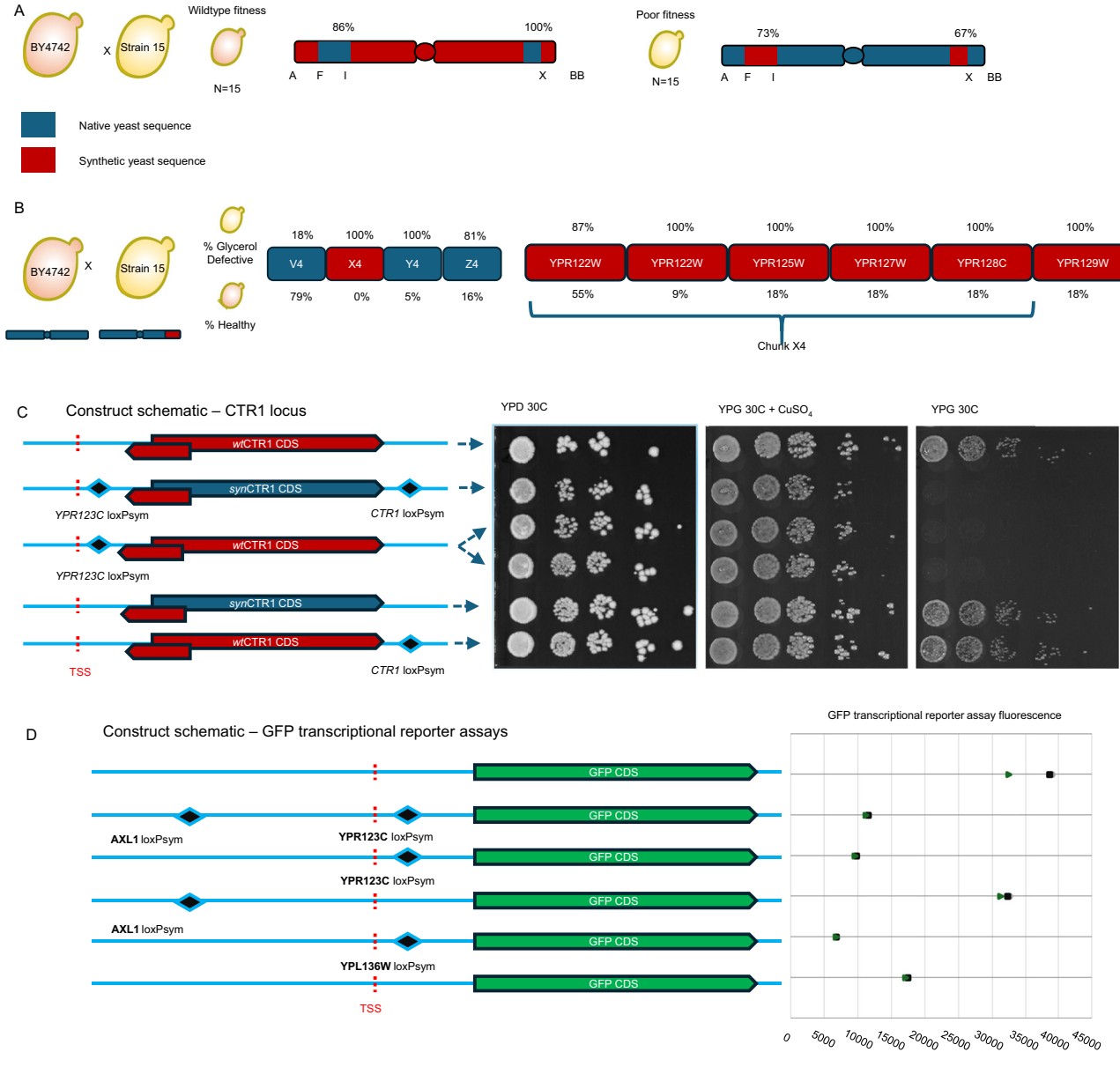

**Fig. 3 | Identification of defective loci using backcrossing and validation using transcriptional reporter assays. A** Synthetic sequence shown is depicted in red and wildtype sequence in blue. From 30 strains, 86% of fit F3 progeny had no synthetic DNA between loci F-I, 100% had no synthetic DNA on chunks X3-X4, while 73% and 63% of F3 unfit progeny had synthetic DNA between megachunks F-I and X3-X4, respectively. **B** Strain 11/BY4742 cross progeny analysis yielded 19 strains with a crossover between megachunks V and Y. Strains bearing synthetic tags for X4 were unfit. A high proportion of strains with synthetic DNA at *AXL1* (*YPR122W*) and *YLH47* (*YPR125W*) showed poor fitness. **C** Schematic diagrams depicting the loxP-sym sites in five strains encoded on BY4741 chromosome XVI, transcription start sites (TSS) shown as a red line and corresponding spot assays on various media to the right. From top to bottom, BY4741, BY4741 with synthetic *CTR1* (*YPR124W*; Strain 29), BY4741 with a *YPR123C* loxPsym in the 5′ UTR of *CTR1* in duplicate (Strain 30), BY4741 with the synthetic coding sequence for *CTR1* (Strain 31), BY4741 with the *CTR1* loxPsym (Strain 32). **D** Six BY4741 strains with GFP constructs in pRS413 with 1 kb upstream from *CTR1* or *GIP3*, and corresponding fluorescence data. Fluorescence data shows the median fluorescence intensity of GFP⁺ gated cells of biological triplicates in mid exponential growth represented by a square, triangle and circle (raw data in Supplementary Fig. 4). From top to bottom, a *GFP* construct with BY4741 *CTR1* 5′UTR sequence with *GFP, CTR1*, and *AXL1* loxPsym sites, the *YPR123C* loxPsym site only, the *AXL1* loxPsym only, the *GIP3* loxPsym, and the BY4741 sequence of *GIP3* followed by *GFP*.

which had similar growth to the parental diploid (Supplementary Fig. 5). Whole genome sequencing identified a potential genetic locus responsible for defective growth on YP-Glycerol at 37 °C.

A single CRISPR D-BUGS colony (Colony 7, Supplementary Fig. 5) exhibited improved growth on YP-Glycerol at 30 °C compared to the parental hybrid diploid and was homozygous for a wild-type PCR tag encoded on *GIP3*. Whole genome sequencing revealed a chromosomal crossover in this strain where *synXVI* sequences were replaced with BY4742 alleles within 30 kb of the

Cas9 cut site (Supplementary Fig. 6). Adjacent to the wild-type PCR tag encoded on *GIP3* in this strain, was a missing loxPsym site downstream of dubious ORF *YPL136W*, situated 3 bp from the 3′ end of *YPL136W*, but 10 bp from the 5′ end of *GIP3* encoded on the complementary strand. A transcriptional reporter assay was conducted by expression of GFP under control of a 1 kb upstream region with, and without the loxPsym site assigned to the dubious ORF overlapping *GIP3, YPL136W*. These reporter assays revealed higher mean fluorescence intensity in cells with a cassette encoding

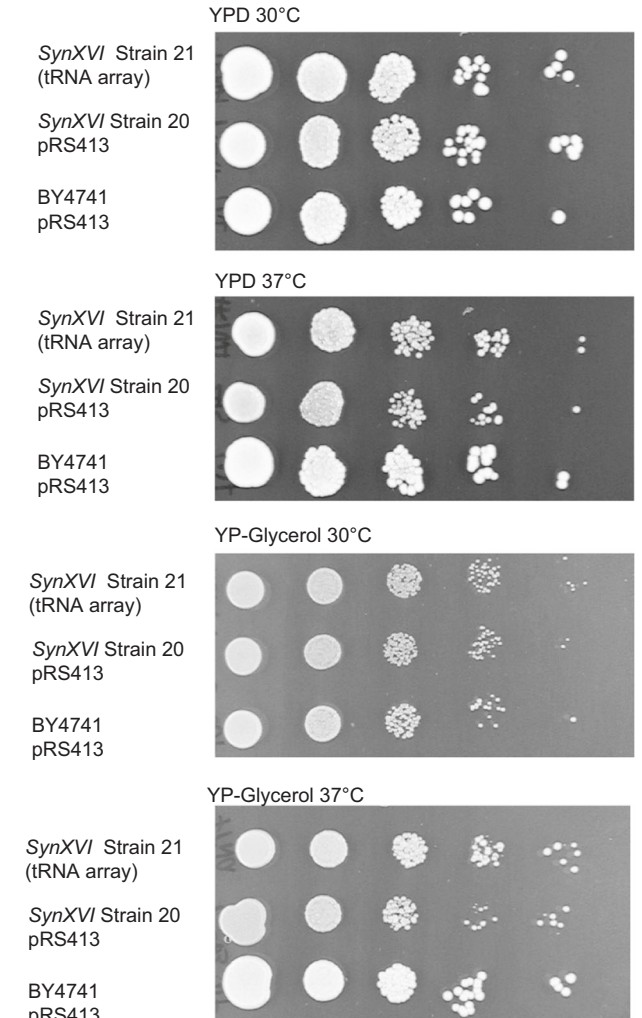

YPD 30°C

*SynXVI* Strain 21 (tRNA array)

*SynXVI* Strain 20 pRS413

BY4741 pRS413

YPD 37°C

*SynXVI* Strain 21 (tRNA array)

*SynXVI* Strain 20 pRS413

BY4741 pRS413

YP-Glycerol 30°C

*SynXVI* Strain 21 (tRNA array)

*SynXVI* Strain 20 pRS413

BY4741 pRS413

YP-Glycerol 37°C

*SynXVI* Strain 21 (tRNA array)

*SynXVI* Strain 20 pRS413

BY4741 pRS413

**Fig. 4 | Effect of pRS413-tRNA array on growth of *synXVI*.** BY4741 and strain 20 harbouring synthetic megachunks A2 to BB4 were transformed with a pRS413 empty vector, and a pRS413 vector harbouring 17 tRNAs removed in the design of chromosome *XVI* (Strains 20 and 21, respectively). Strains demonstrated impaired growth on non-fermentable carbon sources at 37 ˚C. Introduction of the 17 tRNA genes shows improved fitness in the synthetic haploid strain compared to the vector-only control at 37 ˚C, which is especially pronounced on YP-Glycerol medium.

a wild-type *GIP3* 5' UTR before a GFP coding sequence, compared with cell populations with a loxPsym site in the 5' UTR region (Fig. 3D, Supplementary Fig. 4).

### Analysis of a tRNA plasmid
Introduction of both native BY4741 *CTR1* and *GIP3* sequences into the *synXVI* strain 15 (generating Strain 20), improved fitness of the synthetic chromosome strain likely due to the lack of both loxPsym sites assigned to the overlapping dubious ORFs. The improved strain still exhibited impaired growth on glycerol as a carbon source at 37 ˚C.

A design feature of the Sc2.0 synthetic genome included the removal of the highly repetitive tRNA encoding sequences and relocating them to a synthetic neochromosome. Multiple synthetic yeast chromosomes synthesized as a part of the Sc2.0 consortium demonstrated defective growth caused by removal of the tRNA species[19,26]. To rule out deleterious growth phenotypes due to removal of the 17 tRNA species encoded on chromosome XVI (Supplementary Table 2), a pRS413 plasmid containing all corresponding tRNA species from

chromosome XVI was constructed. Strain 20 (with bug fixes in the I and X loci) was transformed with the tRNA array centromeric plasmid (Strain 21). Compared to the wild-type strain with an empty pRS413 vector, and a *synXVI* strain transformed with an empty pRS413 vector, improved growth was observed on introduction of the tRNA array when grown at 37 ˚C on non-fermentable carbon sources (Fig. 4). This final strain, *SynXVI* with the tRNA, exhibiting wildtype-like growth, still bore minor differences to BY4742 when grown on non-fermentable carbon sources at 37 ˚C. Further characterisations were undertaken to explore potential elements contributing to the disparity.

### Characterisation of a loxPsym associated defect
Defective growth caused by both the X and the I loci were caused by insertion of loxPsym sites assigned to overlapping dubious ORFs, and a similar phenotype identified in construction of *synXIV* attributed to the loxPsym sites flanking *MRPL19*, predicted to interrupt mitochondrial RNA targeting motifs. Chromosome *XVI* also encodes a mitochondrial ribosomal protein of the large subunit similar to *MRPL19, MRPL51*. Complementation of *MRPL51* was undertaken to determine whether the loxPsym site associated with it also contributed to poor growth on YP-Glycerol at 37 ˚C. Growth of Strain 21 with wildtype *MRPL51* expressed on a pRS415 vector exhibited improved fitness on when grown on glycerol at 37 ˚C compared to Strain 21 with an empty vector as a control (Supplementary Fig. 7), and when integrated into the chromosome (Supplementary Fig. 9).

### Complementation of individual genes on *synXIV* strain 21
Given multiple lines of evidence that defective growth was linked to loxPsym sites assigned to overlapping dubious ORFs, the *synXVI* chromosome sequence was analysed in-silico to reveal a total of 10 of these motifs in the original design (*UIP4, CTI6, GIP3, ELP4, UBP16, MAK3, NVJ2, MRPL51, CTR1*, and *SGE1*; Supplementary Table 3). To determine whether the expression of these 10 genes was interrupted by loxPsym sites, comparative global proteomic analysis between BY4741 and strain 15 harbouring chunks B to BB was employed.

Global proteomic analyses of these genes did not show a consistent pattern of differential protein abundance between BY4741 and strain 15 proteomic samples (Supplementary Table 4). Since complementation of *MRPL51* demonstrated improved fitness of a strain harbouring *synXVI*, we therefore undertook to evaluate the fitness through complementation of the remaining genes from the list (*UIP4, CTI6, ELP4, UBP16, MAK3, NVJ2*, and *SGE1*) to determine whether they impacted cellular fitness (Supplementary Fig. 8).

The restoration of both the native *CTR1* and *GIP3* 5' UTRs had shown significant improvements in *synXVI* growth, as such, each candidate gene had 1 kb upstream and 0.5 kb downstream of its native ORF cloned into the MCS of an empty pRS415 vector. Once confirmed by Sanger sequencing to be error free, each vector was independently transformed into strain 21 which contained the pRS413 vector with the 17 tRNA species. Growth assays of all seven genes co-introduced with the tRNA array was conducted to evaluate the hypothesis that these genes contribute to defective growth and that individual complementation could improve fitness at elevated temperatures on non-fermentable carbon sources. Marginal improvements were seen in strains with *ELP4, UBP16*, and *NVJ2* (Supplementary Fig. 8).

### Redesign of *synXVI*
Synthetic chromosomes of the Sc2.0 projects have previously reported modifications resulting in defective growth phenotypes[16,20,28]. At the outset of Sc2.0 the *S. cerevisiae* genome included many annotations of ORFs which had yet to be confirmed but have been updated since as dubious ORFs. The initial Sc2.0 project included assignment of loxPsym sites to some of these loci, overlapping the 5' UTR of other genes, causing defects in strains harbouring versions of *synXVI, synII*, and *synXIV*[20,29], amongst others. In light of these lessons, *synXVI* was

## Sc 2.0 SynXVI by Biostudio

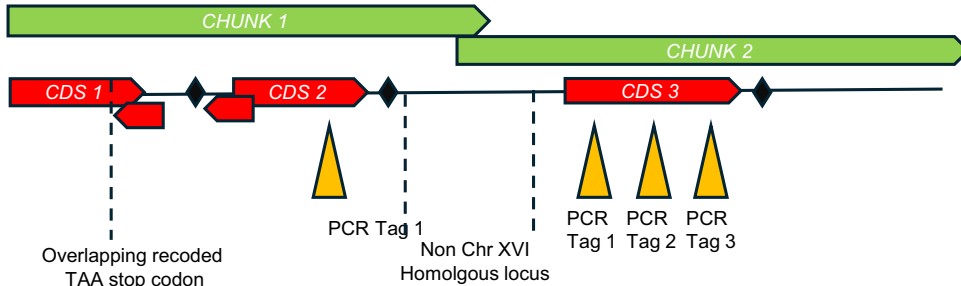

## Sc 2.0 inspired redesign of SynXVI

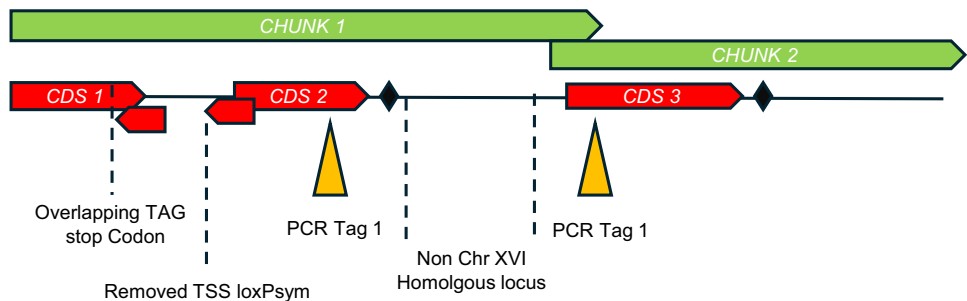

| Design | loxPsym | TAA stop | PCR tags |
|---|---|---|---|
| SynXVI | 330 | 124 | 1208 |
| SynXVI 2.0 | 314 | 118 | 726 |

**Fig. 5 | Potential design feature changes as a result of lessons learned in the design and construction of *Sc2.0*.** The synthetic chromosome *XVI* sequence originally designed in *BioStudio*[15] was modified with *Geneious* software to account for changes in technology and throughput of DNA sequencing. Chunk and megachunk boundaries were moved to avoid possible insertion into non-chromosome XVI loci during construction. LoxPsym sites assigned to dubious ORFs have been removed. ORFs now defined as putative or dubious ORFs have had the TAG → TAA stop codons recoded, when they overlap with coding sequences of other genes. PCR tags were minimised, due to the developments in long read sequencing technologies now ubiquitously available. This redesigned chromosome is included as a supplementary file.

redesigned in-silico using *Geneious Prime (version R9.0)* software, by integrating sequences from BY4741 and Sc2.0 *synXVI* as design templates, to serve as a guide and inspiration for future iterations of Sc2.0 and other eukaryotic genomes. Changes implemented were guided by other findings in the Sc2.0 project and features specific to *synXVI* construction and debugging[16,20,27].

The design of Sc2.0 *synXVI* included 19 megachunks where coding sequences were disrupted by the insertion of either the *URA3* or *LEU2* marker (Supplementary data 1). One example is *SUV3*, an ATP-dependent RNA helicase essential for mitochondrial genome maintenance known to result in increased frequency of mitochondrial Rho⁻ mutants, was disrupted by the *URA3* marker encoded on chunk O5. The location of the marker in the termini of 11 megachunks (C, D, E, F, K, L, N, O, U, V, and AA) ending in coding sequences of genes known to have essentiality for thermotolerance, osmotic stress resistance, respiratory growth or which are known to be involved in increased Rho⁻ mutation frequency could be moved to avoid disruption of the coding sequences. To minimize the potential impact of such occurrences, chunk sizes and termini were modified for each megachunk terminating in genes which impact core cell function. The 40 bp overlap termini of three chunks (A1-A2, C2-C3, D2-D3) with high homology to multiple loci in the yeast genome (such as paralogs) were modified to end in unique chromosome XVI sites to avoid misdirected chunk integration (Fig. 5). The *synXVI* 5′ telomeric chunk A1 was redesigned to account for the

high level of homology to the 3′ telomeric sequence of the BY4741 chromosome *XV* telomere. The original Sc2.0 in-silico changed all stops codons to TAA, along with introducing loxPsym sites in all annotated ORFs, including dubious ORFs overlapping with functional genes. In the redesign recoded TAG-TAA stop codons in dubious ORFs will be reverted to TAG, and all dubious ORF loxPsym sites will be removed, along with loxPsym sites within 250 bp of a start codon of verified ORFs. Two overlapping PCR tags encoded in *PPQ1* on chunk G1 which resulted in an amino-acid change were removed. Redundancy of PCR tags introduced by 'PCRtagger' was reduced to a maximum of one tag per gene per chunk.

The resulting simplified synthetic *synXVI* version 2.0 has 314 loxPsym sites, 118 recoded TAA stop codons, and 726 PCR tags, compared to the original 330 loxPsym sites, 124 recoded TAA stop codons and 1208 PCR tags, and could represent the first step of a second iteration of the chromosome scale design build test learn cycle (Fig. 5).

## Discussion

During construction of *synXVI*, three defects identified were systematically traced to loxPsym sites inserted at the termini of dubious ORFs overlapping verified *S. cerevisiae* genes, indicating a pattern previously found across multiple synthetic Sc2.0 chromosomes affected *synXVI*. Interruption of the 5′ UTR of genes with key physiological functions for yeast, particularly copper metabolism and chromosome segregation

were characterised. *S. cerevisiae's* copper metabolism is key to yeast's unique role in winemaking[30], and due to copper's essentiality for function of key mitochondrial cuproenzymes such as superoxide dismutase (*SOD1*), cytochrome *C* oxidase (*CcO*), and *CcO* assembly proteins Cox11, Cox17 and Sco1, it is an essential trace element for respiratory growth[31]. In excess, copper can be highly toxic and thus import of copper is a tightly regulated cellular process. The import of copper is mediated by *CTR1*, *S. cerevisiae's* primary copper transporter, one of a family of three copper transport genes, *CTR1*, *CTR2* (for vacuolar copper transport), and *CTR3* (a redundant paralogue which is not expressed in BY4741 due to existence of a Ty2 site in the promoter region). *CTR1* deficient mutants are unable to grow on non-fermentable carbon sources due to the crucial role of copper in *CcO*, however with addition of exogenous copper this phenotype is restored[32]. Similarly, strains bearing the loxPsym site associated with *YPR123C* are unable to grow on glycerol as a sole carbon source. Introduction of copper to the media when growing on glycerol as a growth substrate restores growth of the *YPR123C* loxPsym bearing strains (Fig. 3). This demonstrates *CTR1* expression in respiratory growth conditions is improperly induced in strain 15, leading to cell death. Restoration of this genotype improves fitness on glycerol, however with further defective genetic loci across the 1 Mb chromosome such as *MRPL51* and *GIP3* the growth is still less than wildtype fitness.

The altered expression of *CTR1* relative to the presence or absence of loxPsym sites was corroborated with a *GFP* transcriptional reporter assay in BY4741. The introduction of the loxPsym site in the 5' UTR resulted in abnormal median GFP fluorescence (Fig. 3, Supplementary Fig. 4) and is presumably a consequence of the inverted repeat structure of the loxPsym site, which is predicted to from a hairpin structure that could interfere with transcriptional control. A similar feature disturbed production of Glc7-interacting protein 3 (*GIP3*) encoding gene, essential for mitotic spindle formation[33]. In cells where *GIP3* and *GIP4* are overexpressed, the yeast cell's Type 1 S/T protein phosphatase (PP1) catalytic subunit, *GLC7*, is relocalized from the nucleus, interrupting its appropriate function, indicating that Gip3 plays a role in regulating the nuclear functions of Glc7. Glc7 is a multifunctional phosphatase which has dynamic localisation throughout the cell; dysregulation of its function has implications for many core cell functions ranging from mitotic spindle formation, septin formation, meiosis, and as a cell-cycle progression regulator[34]. Gip3 itself is found localised throughout the cytoplasm, ER and mitochondrion, implying possible dynamic localisation, or a role throughout the whole cell[35,36].

During construction of *SynXIV* of the Sc2.0 project, a loxPsym site adjacent to *MRPL19*, a gene encoding a mitochondrial ribosomal protein of the large subunit, was shown to result in mislocalisation of mRNA to the mitochondrial surface, thus was causative of a defect[20]. *MRPL51*, encoded on chromosome *XVI*, also encodes for a mitochondrial ribosomal protein of the large subunit. In the design of both *synXIV's MRPL19* and *synXVI's MRPL51*, annotation of the genome with overlapping ORFs have been automatically assigned loxPsym sites. A rational approach to this problem demonstrated that disruption to this locus by introduction of loxPsym sites did in fact impair mitochondrial function (Supplementary Fig. 7). This led to the conclusion that across *synXVI* systematic insertions of loxPsym sites across the *synXVI* chromosome design has potentially led to aberrant transcription of up to ten genes, as demonstrated for *CTR1*, *GIP3*, and *MRPL51*.

Challenges in SwAP-In[28] based transformation approaches have been improved during the Sc2.0 project particularly by using parallelized approaches for megabase scale chromosomes[16,37,38], but there is still potential for further improvement to the design parameters. The design parameters of Sc2.0 can be used as a guide for future genomes of many other synthetic eukaryotic genomes, drawing inspiration from the 903 kb synthetic chromosome XVI described here as well as the

other chromosomes constructed as a part of this consortium led project. Build errors identified in the construction of *synXVI*, such as duplicate chunks or megachunks or pUC vector insertions, were common to many other chromosomes, one major development since the outset of the Sc2.0 project has been improved global access to long-read sequencing, which in-future could help to rapidly identify clones without these build errors, expediting construction process by removing the need for repair work, and reducing reliance on PCR tags. Rapid construction was facilitated by two main synthetic genomic methodologies. The use of Cas9 to exploit *S. cerevisiae's* naturally high frequency of homologous recombination to remove elements such as chunk duplications and pUC vectors, and previously described parallelized construction protocol, an efficient strategy for chromosome construction[16,37,38]. This allowed large segments of synthetic chromosome to be constructed in different strains in parallel, and subsequently consolidated meiotically. The extreme version of this strategy is to introduce each megachunk individually as a first step. In this way the bugs are identified early in the project and can be eliminated before joining buggy megachunks to adjacent regions[37].

An opportunity to improve the Sc2.0 approach could be to develop new SwAP-In approaches or new ways of inserting large lengths of DNA into the genome to avoid interruption of genes required for optimal growth. As with many of the synthetic chromosomes, the *URA3* and *LEU2* genes in terminal chunks of *synXVI* disrupted several genes, including genes essential for mitochondrial genome maintenance resulting in occasional Rho⁻ mutants[39] (such as *YPL029W* deleted with insertion of the O5 *URA3* marker) which proved detrimental to screening of intermediate *synXVI* strains, presumably because such strains are quite susceptible to damage to mtDNA. Furthermore, insertion of *URA3* or *LEU2* markers adjacent to key genes such as the mitochondrial tyrosyl-tRNA synthetase *MSY1/YPL097W*, may impair expression of those genes, resulting in complex phenotypes and high Rho mutant frequencies which can be technically challenging to debug[40]. Mating mutant strains often introduced healthy mitochondrial DNA back into the impaired cells, while also restoring copies of previous marker interrupted genes. While yeast is capable of growth without functional mitochondrial genomes, in the absence of selective pressure for functioning respiratory growth in Rho⁻ mutants there is a risk of accumulation of deleterious mutations in genes relating to respiratory growth. In some cases, these mitochondrial defects affect diploid strains' ability to successfully sporulate, a critical step in parallel construction approaches. It is therefore important to maintain high fitness and avoid generation of Rho⁻ mutants in the construction of yeast genomes specifically. This lesson could also be applied to important conditionally essential physiological functions when building non-yeast genomes, such as photosynthesis in *Chlamydomonas reinahrdtii*[41,42] or other green lineage organisms. Furthermore, the SwAP-In based methodology, while effective at inserting large segments of DNA to construct chromosomes and genomes in yeast[28], may have limits in organisms with low rates of homologous recombination or do not have adequate tools to improve homologous recombination rates such as established Cas9 based methodologies. Development of plasmid technologies in some non-yeast species such as marine diatoms[43], and genetic improvement of homologous recombination rates in other species[44] are examples of current steps showing promise of extending the vision of Yeast 2.0 to new organisms such as plants[45] and algae[46]. The existence of an exhaustive database on gene function and essentiality such as the Saccharomyces Genome Database, SGD[47], was crucial for whole genome design in Sc2.0, and few other organisms sport an equivalent resource with the same depth of annotation and links to raw data. In lieu of such resources for other organisms, at the very least, application of high-throughput genetic screens to identify genes essential for core physiological functions[48] should be employed before whole genome design approaches are undertaken.

*Proposed* in-silico redesigns have been developed for the next iteration of the Sc2.0 project, incorporating lessons learned from the global Sc2.0 project. Alternatively, design and construction of neo-chromosomes incorporating native sequences for all genes with 5' UTRs interrupted by loxPsym sites inserted in overlapping dubious ORFs across the entire yeast genome could be considered worthwhile. This is an important lesson to be noted in future synthetic eukaryote genome designs[46], while *S. cerevisiae* can survive mitochondrial genome loss under permissive conditions, and major cellular perturbations, more complex organisms may not be so robust and could benefit from iterations of the design build-test-learn-cycle incorporating fewer changes in the initial design stages. Moreover, essential, and conditionally essential genes should be defined in advance.

Various synthetic chromosome strains have reported defects due to the removal of tRNA genes. The original design concept accounted for this, and commissioned the construction of a supplemental tRNA neochromosome[16,19,20,29], allowing for reintroduction of all removed tRNA genes. In the nuclear genome of *S. cerevisiae*, tRNA molecules are highly redundant; however, the impact of changes to isoacceptor tRNA frequencies remains unclear. Reintroduction of synthetic versions of the 17 tRNA molecules which had been removed from the *synXVI* chromosome on a centromeric plasmid vector resulted in improvements in cellular fitness. *S. cerevisiae* exhibits high levels of tRNA redundancy.

Removal of all tRNA molecules in *S. cerevisiae* as designed in the Sc2.0 project would render a cell inviable. Individual knockouts of tRNAs with lower than average redundancy have been shown in some cases to have no impact on overall cell fitness, as is the case with the removal of one of the cell's four initiation methionine tRNAs (tM(CAU)P) encoded on chromosome *XVI* (and subsequent inclusion on the tRNA neochromosome[49]). Construction of *synIII* demonstrated that despite removal of a single copy tRNA tQ(UUG)C, the cell harbouring *synIII* exhibited high levels of chromosomal stability[50]. A similar phenomenon was noted during construction of *synXII* with two copies of tL(UAG) growth phenotypes were restored with complementation. Therefore, it was not expected that the stepwise chromosome by chromosome approach would impact cell fitness given tRNA redundancy. Nevertheless, there are low levels of redundancy of certain tRNA species on chromosome XVI such as tS(UGA) tRNAs which has three copies and tC(GCA) tRNAs which has four copies in BY4741, resulting in a strain with only two copies of each. The low redundancy of certain tRNAs on *synXVI* (Supplementary Table 2) may have impacted cell fitness combinatorially, consistent with previous reports that tRNA abundance is relative to the frequency of those codons in a genome[49,51]. Ultimately this demonstrates the utility of the tRNA neochromosome project[19] to the viability and fitness of genome scale recoding of eukaryotes. It is worth noting that the removal of particular codons has been demonstrated as an extremely worthwhile biocontainment barrier, in conjunction with a requirement for exogenous non-natural nucleotides, and elimination of stop and sense codons has been demonstrated to confer viral resistance to organisms[52,53]. The whole genome approach of modifying the tRNA and codon complement of Sc2.0 may represent the first step of a future biologically isolated yeast cell.

To guide future approaches to synthetic genome design, the lessons from the *synXVI* and other consortium chromosomes were used to redesign a debugged blueprint of the *synXVI* chromosome. Integration of multiple PCR tags have contributed to defective growth phenotypes during the Sc2.0 project[16,28,54], and an error from the *BioStudio* PCR-Tagger resulting in two overlapping PCR tags modifying the peptide sequence of *YPL179W*, resulting in a S119 → R119 mutation in a serine-serine-serine motif, have demonstrated potential value in taking advantage of technological advances allowing for coding-sequence conservative designs without sacrificing non-coding sequence modifications such as loxPsym insertions and intron removals. The Sc2.0 project was developed at a stage where long-read sequencing was nascent. The cost of screening was faster and cheaper when done employing PCR or qPCR based tag screening strategies. In recent years particularly with the advent of cheap nanopore sequencing technology, alternative strategies can give rapid extensive genotype data whilst mitigating the potential for disruption to coding sequences. This permits a more conservative design approach whilst providing a means to rapidly confirm insertion of all designed features.

At the outset of the Sc2.0 project, the available S288C genome sequence included multiple putative ORFs which have since been recognised as unlikely to code for functional genes. In the iterated genome design, loxPsym sites assigned to overlapping dubious ORFs were removed, recoded stop-codons of dubious ORFs in coding sequences of overlapping genes were reverted. This lesson, applied to non-*Saccharomyces* genomes highlights the relevance of high quality, well annotated starting genome sequences from which to base synthetic genome designs.

Alternate terminal chunk loci have been proposed where all terminal chunks are inserted away from coding sequences of genes involved in mitochondrial genome maintenance or respiratory growth and a chunk design that allows for chromosomal specificity for the subtelomeric homology between chromosome XVIL telomere and chromosome XVR telomere. Chunks with termini with high sequence similarity to non-chromosome XVI target loci may insert incorrectly. These were redesigned, however future designs may benefit from terminating chunks at unique sequences only, potentially including recoded PCR tags. While these steps may improve future designs, some elements may need to be uncovered experimentally through the classic design-build-test-learn cycle. Alternatively, adaptation to novel genetic configurations can be achieved through exploitation of the natural plasticity of genetic systems as with *synXIV* through processes such as adaptive laboratory evolution[20]. Overall, many elements of the Sc2.0 project have helped achieve a high level of biological predictability central to the thesis of synthetic biology[55] such as removal of the long terminal repeats, as potential source of random recombination[15]. These approaches mark steps towards bringing iterative biological design to a whole genome scale and may have an important impact on the future of synthetic biology[56].

For future iterations of synthetic yeast genome design and construction from a broader perspective it is worth considering whether alternative strains may suit industrial exploitation more readily. Strains derived from S288C which have a higher propensity to generate rho⁻ phenotypes, and with, for example, no functional *CTR3* gene, future endeavours may produce more industrially relevant organisms if they are based on the genome of more robust strains such as strain W303[57]. Design of genomes could be inspired by other genomes found in nature which are more given to redesign for human purposes as opposed to natural evolution. Genomes which have many uniformly sized chromosomes such as seen in the genome of *Galdieria sulphuraria*. *G. sulphuraria* is a thermophilic microalga which has a genome with a 13.1 Mb genome spread across 72-73 chromosomes[46] which would be easier and faster to construct or synthesize rapidly. More importantly, a major challenge unique to the six large chromosomes of Sc2.0 including *synXVI* has been the occurrence of multiple defects on the same chromosomes, which are difficult to trace even with CRISPR D-BUGS and classic breeding. By proposing similar genomic content as Sc2.0 but across many small chromosomes, occurrence of defect causing genetic material can be more easily traced.

Despite challenges during the Sc2.0 project, *S. cerevisiae* continues to demonstrate its utility as a host for eukaryotic chromosome and genome scale projects. From its early use in design and synthesis of a bacterial genome[58], construction of the yeast pan-genome neochromosome[5] to elaborate human artificial chromosome design, construction and streamlined spheroplast based transfection into mammalian cells, yeast's unique genetic malleability offers a fantastic

platform for synthetic chromosomes. As techniques such as CRISPR-DBUGS useful for debugging phenotypically deleterious loci[27], CReATiNG for rapidly generating yeast chromsomes[59] emerge, we can expect to see broad reaching outcomes for agricultural and therapeutic applications, such as karyotype engineering[22] for agriculturally relevant species and new techniques such as KaryoCreate[60] for engineering mammalian cells. We believe these works, this project and the broader Sc2.0 consortium demonstrates that *S. cerevisiae* holds great potential in libraries of new chromosomes, neochromosomes, and genomes for subsequent transfer into cells to create life forms tailored to humankinds needs.

The lessons from the assembly and debugging of the 902,994-bp synthetic *S. cerevisiae* chromosome *synXVI* of the Sc2.0 has been described. With pooled-segregant whole-genome analyses and CRISPR D-BUGS methods, all fitness related synthetic modifications were identified and characterised. Defective loci caused by systematic insertion of loxPsym sites to overlapping dubious ORFs were repaired by removal of the loxPsym sites from affected 5′ UTRs of key genes. Reintroduction of low-redundancy tRNAs along with the rest of the tRNA molecules from chromosome XVI improved fitness of the *synXVI* strain.

## Methods

### Culture conditions
Strains used in this study can be found in Supplementary data 1. Strains in this project are derivatives of BY4741 (MATa *his3Δ1 leu2Δ0 met15Δ0 ura3Δ0*) and BY4742 (MATα *his3Δ1 leu2Δ0 lys2Δ0 ura3Δ0*). Unless otherwise stated standard yeast cultivation was undertaken at 30 ˚C with shaking at 200 rpm in an orbital shaker. For routine culturing YPD medium (1% yeast extract, 2% peptone and 2% dextrose) was used. Solid media were prepared with 2% agar. Synthetic Dextrose medium (SD) was prepared with 2% dextrose, 0.67% Yeast Nitrogen Base, and amino acids as needed at 100 mg/L final concentration. Dropout media (SC⁻Ura, SC⁻Leu, etc.) were prepared according to Sherman[61]. *Escherichia coli* cultivation was undertaken at 37 ˚C with shaking at 200 rpm using Lysogeny broth with appropriate ampicillin (100 μg/mL) or chloramphenicol (50 μg/mL) dependent on bacterial vector. Cultures were undertaken in volumes of 10 mL in 5 mL falcon tubes or 50 mL baffled glass flasks.

### Megachunk preparation
Synthetic yeast DNA was synthesized (Genscript) and supplied in cloned pUC18 vectors as designed by open source software[15] (BioStudio). All vectors and constructs used this study can be found in Supplementary data 1. Each DNA megachunk consisted of four to five DNA chunks, with terminal chunks (except for the telomeric megachunk BB4) encoding either *URA3* or *LEU2* as auxotrophic markers. A 450 bp overlapping region with homology to flanking megachunks was included as well as restriction sites that served as construction boundaries. Chunk DNA was transformed into *E. coli* DH5a, grown overnight, and extracted using Monarch miniprep kits (New England Biolabs). Chunks were prepared either by PCR amplification using oligonucleotides designed with annealing temperatures of 60 ˚C with 40 bp overlaps to adjacent chunks, or by restriction digest. For PCR amplicon transformation, DNA was amplified with Phusion Polymerase using high-fidelity enzyme buffer (New England Biolabs), treated with DpnI restriction endonuclease (New England Biolabs) for 1 hour at 37 ˚C, and purified using a Monarch PCR purification kit (New England Biolabs), quantified, and subsequently transformed in at a 1:1 molar ratio for each chunk. For restriction digest DNA was incubated with corresponding enzymes for 2 hours at temperatures recommended by the supplier, then extracted using a Monarch gel extraction kit (New England Biolabs). Gel extracted DNA was combined in a molar ratio 4:2:1:0.5, with at least 200 ng of terminal chunk DNA bearing the selective marker (chunk 1, 2, 3, 4, where chunk 4 encoded *URA3* or

*LEU2*), ligated at 16 ˚C overnight with T4 ligase (New England Biolabs) and transformed into yeast.

### Yeast transformation
Yeast strains were precultured overnight in 10 mL YPD and subsequently used to inoculate 20 mL of YPD at OD 0.125. Once OD reached 0.5, cells were pelleted, washed with sterile MilliQ water, and brought to 0.1 M lithium acetate with 50% polyethylene glycol to a final volume of 360 μL with 10 μL of herring sperm DNA (Sigma), and transforming DNA. Cells were then incubated at 30 ˚C for 30 minutes then 42 ˚C for 45 minutes. Cells were then pelleted, resuspended in 400 μL of sterile MilliQ water, and plated on four petri dishes under appropriate selection conditions.

### Megachunk SwAP-In
Two to three days post transformation individual colonies were picked and patched onto YPD agar plates as a positive growth control and SC medium lacking either leucine or uracil, depending on the corresponding megachunk marker. Cells were observed the following day for positive growth on YPD but lack of growth on uracil or leucine deficient plates. These cells were selected for qPCR-based tag analysis to confirm the presence of synthetic DNA in the appropriate loci using oligonucleotides detailed in Ssupplementary data 1. Once confirmed, strains were propagated in YPD media overnight, and stored as cryogenic stocks with 20% glycerol at −80 ˚C.

### Fitness assessment
Yeast fitness was assessed by spot assay. Briefly, yeast cells were precultured overnight, precultures were used to inoculate cultures at $OD_{600}$ 0.125, which were allowed to grow for two divisions to reach $OD_{600}$ 0.5. Cultures at $OD_{600}$ 0.5 were serially diluted 1:10 in sterile MilliQ water and spotted on YPD and YP-Glycerol (1% yeast extract, 2% dextrose and 2% filter sterilized glycerol) or SC medium where required. Plates were grown at 30 ˚C and at 37 ˚C for 3-5 days. Cells were then photographed and visualised for colony size and measures of fitness.

### Mating type analysis
Mating type determination was performed by PCR analysis. Genomic DNA was extracted using a rapid DNA extraction method[62] followed by fragment amplification using GoTAQ green 2x mastermix as recommended by the manufacturer[43]. The universal *Mat* locus R oligo (5′AGTCACATCAAGATCGTTTATGG 3′) was used in combination with the *MATa* locus specific oligo, Mat-A_forward (5′ ACTCCACTTCAAGTAAGAGTTTG 3′) or the MATα locus specific, Mat-alpha_forward (5′ GCACGGAATATGGGACTACTTCG 3′).

### Diploid formation and sporulation
Mating between strains of opposite mating types were promoted by coincubation at a 1:1 ratio in YPD medium at 30 ˚C for 24 hours. Cells were then washed with sterile MilliQ water and plated either on selective media or streaked for single colonies. Diploids were validated either with mating type PCR analysis or with qPCR amplification for heterozygosity at relevant PCR tags on the synthetic chromosome and or tested for mating type alleles using BY4741 and BY4742 as controls.

Sporulation was conducted by incubating diploids on solid sporulation plates at 30 ˚C. Sporulation plates consist of 2% bacteriological agar, 0.5% yeast extract, 1% potassium acetate, and 0.005% zinc acetate. Samples were taken from sporulation plates and examined under microscopes to identify tetrads after three days of incubation. Recuperation of cells was by rinsing with 1 mL MilliQ water.

### RT-qPCR tag analysis
RT-qPCR was conducted on cells which had lost or gained an auxotrophic marker through either Cas9 mediated transformation, or

SwAP-In. DNA was extracted from colonies using two protocols. Colonies were picked and suspended in 100 μL of 20 mM sodium hydroxide, incubated for 5 minutes at 95 °C, vortexed briefly, and spun for 1 minute. Supernatant was used as crude DNA extract for PCR tag analysis. Alternatively, 100 mM lithium acetate 50% SDS was used for DNA extraction, and crude DNA extract was purified using ethanol precipitation[44]. PCR was conducted in white Roche qPCR plates in a final volume of 3 μL. PCR plates were filled with 1.5 μL of 2x SYBR green mastermix (BioRad) and 1.2 μL of water per well. Genomic DNA and 15 μM forward/reverse oligonucleotide mixes were transferred using a Beckman Coulter Echo550 acoustic liquid handling robot. 200 nL of crude genomic DNA was transferred per well with 50 nL of each primer pair (per synthetic or wild-type PCR tag). Plates were centrifuged post-transfer. PCR was conducted using a touchdown PCR protocol in a LightCycler 480II (Roche Diagnostics), following a protocol of 95 °C denaturation of three minutes followed by 15 cycles of 30 s at 95 °C, 30 s at 70 °C with a ramp rate of −1 °C each cycle, and extension at 72 °C for 30 s, followed by 20 cycles of constant annealing temperature at 55 °C. SYBR green fluorescence was measured by the Light-Cycler at the end of each extension step. A melt curve was determined with fluorescence measurements every 5 s by heating from 50 °C to 95 °C. Tag PCR amplification curves were compared to wild-type and synthetic control DNA samples to identify strain genotype. The sequence of the oligonucleotides used in RT-qPCR tag analysis is included in Supplementary data 1.

## Spot assay fitness analysis
Spot assays were conducted by preculture of yeast strains in selective medium overnight, cells were diluted to $OD_{600}$ of 0.2 and allowed to grow for four hours at 30 °C in YPD. Cells were transferred to 96 well plates and diluted 10 μL:90 μL in sterile MilliQ water. Serial dilutions were made to produce spots which had single colonies, at 1:10, 1:100, 1:1000 and 1:10,000. Strains were allowed to grow until clear single colonies formed, between 2 and 7 days (depending on growth conditions and media). Plates were imaged using PhenoBooth+ (Singer Instruments) and processed with Microsoft PowerPoint.

## Flow cytometry
Flow cytometry was conducted using cells grown in SD media with or without nutrient supplementation depending on strain auxotrophies. Cells were grown overnight in selective media and were diluted to $OD_{600}$ 0.05 into selective media for four hours. Cells were washed in MilliQ, and data was collected on 50,000 cells per experiment, gating strategies were employed to filter out detritus and dead cells. Histogram gating for determination of fluorescent and non-fluorescent cell populations was performed using FlowJo (Becton, Dickinson & Company), BY4741 cells containing only empty vectors were used as control samples.

## Genome sequencing
Genomic DNA was extracted using ThermoFisher Yeast Genomic DNA Isolation Kit (ThermoFisher catalogue number 78870). Briefly, cells were grown overnight, 10 mL of culture was pelleted, genomic DNA was isolated as recommended by the manufacturer. NGS paired end Illumina sequencing was performed at either Macrogen Inc. (Seoul, South Korea) or Genewiz (South Plainfield, NJ, USA). Genomic reads were mapped to either a BY4741 reference genome sequence or in-silico designed synthetic chromosomes, using Geneious Prime mapper software.

A barcoded library was prepared with purified DNA using a RBK004 rapid barcoding kit (Oxford Nanopore) using the manufacturer's instructions. MinION Mk1B was operated as per protocol outlined by ONT with Spot-ON Flow Cell, R9 Version (FLO-MIN106D) and Flow Cell Priming Kit (EXP-FLP002). MinKnow version 127.0.0.1 was run with real time base calling switched off during sequencing to maximise efficiency of the run. Sequencing was run until approximately 40-fold coverage was achieved. Sequencing was manually halted after an estimated ~50 fold coverage of the yeast genomes that were barcoded was completed.

The raw fast5 datafiles were basecalled with the High Accuracy basecalling algorithm using the Guppy GPU basecaller and barcodes were automatically sorted and trimmed. The fastq files for each barcode were combined into a single file and assembled into contigs with Canu[63]. The assembled genomes usually contained single contigs for each chromosome. The Canu assembled genome sequence was further polished using three rounds of medaka (https://github.com/nanoporetech/medaka). These were compared to the parental strain reference sequence with QUAST[64]. Minimap2[65] was used to map raw reads back to the Canu assembled genome and Samtools[66] was used to create a sorted and indexed bam file for additional viewing with IGV and Geneious software. Raw reads were also mapped against the expected parental strain sequence using minimap2. A manual list was formed of retained stop codons wildtype PCR tags and loxPsym sites (supplementary data 1).

## Rectifying build errors
**Duplication of chunk B2.** In order to rectify a duplication of chunk B2, a *KanMX* marker was amplified from the *S. cerevisiae* gene deletion library and inserted at the locus of *SAM3* in a BY4742 strain. This strain was then used to rebuild, via SwAP-In megachunks A and B. An A-B strain with a wild-type *YPL254C* PCR tag (tag 1 of 2) in the locus of B3, 3' of the duplication, was utilised to track the presence or absence of the duplication. A *synXVI* strain including synthetic megachunks A through to K was transformed with a *URA3* construct targeting the A5 locus. The two strains, A-K *(URA3)* and A-B *(LEU2)* were mated, spores were obtained by random spore isolation, and replica-plated onto YPD, SC[-Leu], and Sc[-Ura]. Colonies which presented as both *ura3Δ* or *leu2Δ* were then selected for genetic screening. Leucine, uracil auxotrophs were then sent for whole genome sequencing. The read coverage at the locus of B2 was equal to the mean read coverage of the *Saccharomyces* genome.

**Missing stop codon on chunk J2.** A stop codon missing in chunk J2 was replaced by transforming the strain with a *URA3* cassette. The J2 chunk was then amplified and co-transformed with p*URA3*-crRNA-cas9-pRS423, a pRS423 vector for expression of Cas9 and a PAM sequence targeting the *URA3* locus. *Ura3Δ* strains were then evaluated for re-insertion of the synthetic DNA by PCR amplicon polymorphism, confirming the *URA3* locus had been replaced by the synthetic copy of the gene including the TAA stop codon.

**Multiple insertions of chunk M3.** A quadruplication of the M3 locus was removed by transformation with a Cas9 CRISPR construct targeting a PAM site unique to the non-native M3::pUC19::M3 junction. Cells were screened for the non-native 3'-5' M3 M3 junction using PCR, removal of the quadruplication was confirmed using nanopore long-read sequencing.

**Removal of KanMX marker and duplication on O3.** O3 repair was performed in a two-step process of insertion of a *URA3* cassette on the chunk replacing the duplicated locus, followed by p*URA3*-crRNA-cas9-pRS423 mediated homology directed repair to insert synthetic O3 DNA in the locus of the O3 *URA3* cassette.

**Insertion of mega chunk O.** Chunk O5, containing the 5' *URA3* marker, was inserted onto Strain 15. The inserted *URA3* marker disrupted the *YPLO29W* ORF, creating the intermediate strain P-W_*YPLO29W*::*URA3* strain. PCR fragments of chunks O1 – O4 and P1 were prepared from the chunk cloning vectors, and terminal-binding primers. Chunk O5 was amplified to exclude the 5'-end *URA3* marker. All six fragments

were combined in equal molar ratios and transformed into the P-W_*YPLO29W::URA3* strain. Putative transformants were selected on 5-FOA agar plates[67]. PCR-Tag analysis validated insertion of chunks O4, O5, and partial insertion of O3 between tags *YPLO39W* amp1v1 and *YPLO29W* amp1v1 (Strain 17).

To provide a suitable region for chromosome arm consolidation, Chunk O1-O3 was added to Strain 8 using CRISPR/Cas9. A dual-vector system, encoding the Cas9 (from the p414-TEF-Cas9) and chunk O2 targeting guide RNA gene (from the p426-SNR52p-gRNA.pUC18) was transformed into Strain 8. The colony with the most complete insertion, contained all recoded DNA except for the two small regions of *YPLO4OC*_1861 and *YPLO4OC*_2137. The two sections of Megachunk O were subsequently combined during the chromosome arm consolidation, discussed elsewhere.

**Insertion of missing PCR-tags on chunk P1.** The *YPLO29W*_897, *YPLO29W*_1191, and *YPLO29W*_1497 PCR-tags were revealed by whole genome sequencing to be missing from Strain 10. A CRISPR/Cas9 approach was used to reinsert the three missing tags within the *YPLO29W* gene. A dual-vector system, encoding the Cas9 (from the p414-TEF-Cas9) and chunk P1 targeting guide RNA gene (from the p426-SNR52p-gRNA.pUC18) was transformed into Strain 10. A repair DNA fragment, generated by PCR from the pUC18_chunk-P1, which included a region with all three recoded tags, was used as a repair template.

**Removal of pUC18 vector between chunk T2 and T3.** Genome sequencing revealed a pUC18 cloning vector insert between chunk T2 and T3. A CRISPR/Cas9 approach was used to remove the unwanted pUC18 DNA. In short, Strain 12 was transformed with a dual-vector system, encoding the Cas9 (from the p414-TEF-Cas9) and pUC18 targeting guide RNA gene (from the p426-SNR52p-gRNA.pUC18), respectively. Two additional overlapping DNA fragments, generated by PCR, consisting of a 119 bp 3′-fragment of the T2 chunk and a 117 bp 5′-fragment from the T3 chunk, were transformed to serve as repair template.

**Removal of pUC18 vector between chunk BB3 and BB4.** A pUC vector inserted between chunk BB3 and BB4 was removed by transformation of a pRS413 vector encoding Cas9 and a PAM sequence targeting an NGG motif at the unintended junction between BB3 and the pUC vector. Cas9 was employed to initiate a double strand break which mediated homologous crossover of the BB3 40 bp homology sequence to BB4, 5′ of the pUC vector, with the BB4 40 bp homology sequence to BB3 3′ of the pUC vector. 48 colonies were screened, and three colonies were identified with an amplicon from BB3 3′ to BB4 5′ equivalent to the normal junction size demonstrating removal of the pUC vector.

**Insertion of missing chunk A1.** A1 insertion into strain 20 was performed in a two-step process of insertion of a *URA3* cassette on chunk A2 targeting the sequence lacking homology to the A1 sub telomeric locus, followed by p*URA3*-crRNA-cas9-pRS423 mediated homology directed repair to insert synthetic A1 and A2 DNA in the locus of the A2 *URA3* cassette, PCR verification of a loxPsym site and a PCR tag on A1 confirmed insertion of both chunks on uracil auxotrophic strains identified by replica plating, yielding Strain 23.

**Consolidation of chromosome arms.** Targeted backcrosses were mediated by integration of combinations of auxotrophic makers, such as *URA3* and *LEU2* at the boundaries of crossover points. AG included a G4 *URA3* marker, while G-J included a H4 *LEU2* marker. A-J included a J4 *LEU2* marker, while J-O included a I4 *URA3* marker. Strains were crossed using random spore isolation and replica plating was used to identify strains which were auxotrophic for uracil and leucine which had crossed in the appropriate locus. Strains were then analysed by qPCR tag analysis and whole genome sequencing.

A final A-BB strain was generated by crossing an A-O strain with an O-BB strain, strains which were *ura3Δ leu2Δ* were selected and analysed for synthetic PCR tags. The most complete strain was chosen for growth defect identification.

**CRISPR D-BUGS.** The CRISPR D-BUGS[27] protocol was applied to identify specific loci responsible for defective growth in the strain harbouring *synXVI*. 24 constructs were built using golden gate assembly in pYZ555 with guide RNA PAM motifs targeting a wild-type sequence corresponding to a PCR tag on the third chunk of every megachunk, in *E. coli*. Two diploid strains (Strains 27 and 28) were generated by mating Strain 15 (with synthetic megachunks from B3 to BB4), and Strains 24 and 25 (BY4742 transformed with *URA3* in the equivalent loci of A2 and BB4), respectively. Strain 27, with *URA3* in the equivalent locus of A2, was transformed with pYZ555 vectors with sgRNA sequences targeting PCR tags on the third chunk of every megachunk from megachunk A to megachunk O, where Strain 28 was transformed with pYZ555 vectors with sgRNA sequences targeting PCR tags on the third chunk of every megachunk from megachunk P to megachunk BB. CRISPR D-bugs vectors were constructed by designing 20 bp PAM sites adjacent to NGG motifs coinciding with BY4741 sequences corresponding to synthetic PCR tags, with flanking BsmBI sites suitable for incorporation into pYZ555, the constructs were ordered as oligonucleotides (IDT) and resuspended in MilliQ water, mixed at a ratio of 1:1, boiled at 95 °C for 5 minutes and gradually reduced to room temperature over half an hour. The PAM sequences were assembled into pYZ555 as per instructions in NEBridge® Golden Gate Assembly Kit (BsmBI-v2) (New England Biolabs). Assembly mix was transformed into *E. coli* with kanamycin (50 μg/mL) and examined under blue light. White colonies were picked, plasmid DNA was extracted and sequenced to confirm insertion of the PAM site into the sgRNA locus. The correct plasmids were then transformed into Strain 27 and Strain 28 as appropriate. Induction of Cas9 was conducted by overnight incubation of cells in YP medium with 2% galactose, cells were diluted 2 μL in 1000 μL MilliQ water and plated on SC media with 5-FOA. CRISPR D-BUGS was conducted for plasmids with PAM sites targeting megachunks A, C, D, E, F, G, H, I, J, K, L, M, N, V, W, X, Y, and Z. At least four independent colonies from each CRISPR D-BUGS transformation were analysed for growth at 30 °C and 37 °C with glucose or glycerol to test for thermotolerance and respiratory growth, with control Strains 27 or 28 (the hybrid synthetic diploids with *URA3* in equivalent locus A2 and BB2, respectively), BY4741 and 15 (the haploid strain with synthetic megachunks B3 to BB4). Additional colonies were analysed for experiments which revealed phenotypic differences between transformants.

**Proteomic preparation and analysis**

**Preparation of Samples for LC-MS/MS.** Protein concentrations in lysates were determined using BCA assays (Pierce, A55864), after which 200–500 μL digestion buffer (1% sodium deoxycholate in 100 mM TEAB) was added. Aliquots containing 20–50 μg total protein were then removed and reduced by addition of DTT to a final concentration of 10 mM for 30 minutes at 60 °C and alkylated with iodoacetamide at a final concentration of 20 mM for 30 minutes in the dark at room temperature. Trypsin (Merck, T6567) was added at ratios of ~50:1 (w/w) and digestions carried out overnight at 37 °C. Sodium deoxycholate was precipitated by addition of 1% formic acid and samples centrifuged for 5 minutes at 14,000 g. Supernatants were transferred to new tubes and evaporated to dryness in a vacuum concentrator, after which peptides were resuspended in 0.1% formic acid.

**LC-MS/MS Data Collection.** LC-MS/MS data was collected using Data Independent Acquisition (DIA) on a Q-Exactive HF-X mass spectrometer (Thermo Fisher Scientific) interfaced with an UltiMate 3000 UHPLC (ThermoFisher Scientific). Peptide samples were injected onto the peptide trap column and washed with loading buffer for 10 minutes. The peptide trap was then switched in line with an in-housed packed analytical nano-LC column (ReproSil-Pur 120 C18-AQ, 3 µm, 250 ×0.075 mm, 75 µm x 30 cm). Peptides were eluted from the trap onto the nano-LC column and separated with a linear gradient of 2% mobile phase B to 35% mobile phase B over 90 min at a flow rate of 300 nL/min, then held at 95% mobile phase B for 5 minutes prior to re-equilibration.

The column eluant was directed into the ionization source of the mass spectrometer operating in positive ion mode. Peptide precursors from 350 to 1,650 m/z were scanned at 60 k resolution with an AGC target value of $3 \times 10^6$, and a maximum injection time of 50 ms. Ions across a series of pre-defined *m/z* windows were fragmented by HCD using a normalized collision energy of 27.5. MS2 scan resolution was set at 30 k, and an AGC target value of $2 \times 10^5$ was used.

**DIA-NN Analysis of DIA LC-MS/MS Data.** DIA LC-MS/MS raw files were analysed using DIA-NN (version 1.8.1). Library-free searches were performed using an in silico spectral library produced from annotated *S. cerevisiae* sequences in the UniProt database (release 2023_03; 6060 sequence entries). The following parameters were employed: minimum and maximum fragment *m/z* set to 200 and 1800 respectively; minimum and maximum precursor *m/z* set to 300 and 1800 respectively; minimum and maximum peptide length set to 7 and 30, respectively; N-terminal methionine excision enabled; protease digestion set to trypsin (fully specific) with a maximum of one missed cleavage; cysteine carbamidomethylation set as a fixed modification; minimum and maximum precursor charge set to 1 and 4, respectively; protein inference based on genes; neural network classifier set to single-pass mode; quantification strategy set to robust LC (high precision); cross-run normalisation set to RT-dependent; library generation set to smart profiling; and false discovery rate threshold set to 1%.

**Statistical Analysis of DIA-NN Outputs.** Individual protein abundances were compared between strain 15 versus BY4741 samples. For pairwise comparisons, only proteins quantified by DIA-NN across all three samples were subjected to statistical analysis. To assess whether proteins were significantly different in abundance, two tailed homoscedastic Student's t-tests were performed using LFQ values. Log transformed (base 2) protein fold changes were also calculated using mean LFQ values for individual proteins.

**Materials availability.** All plasmids and yeast strains generated during this study are available on request.

### Reporting summary
Further information on research design is available in the Nature Portfolio Reporting Summary linked to this article.

## Data availability
The raw sequencing data and genome assembly have been deposited in the National Center for Biotechnology Information (NCBI) under BioProject PRJNA1164922. The proteomic analyses data generated in this study are provided in the Source Data file and are the mass spectrometry proteomics data have been deposited to the ProteomeXchange Consortium via the PRIDE[68] partner repository with the dataset identifier PXD058199. The growth curve data generated in this study are provided in the Supplementary Information/Source Data file. The iterated redesign of synXVI is available as a supplementary file

accompanying this article. Any additional information required to reanalyse the data reported in this paper is available from the lead contact upon request.Further information and requests for resources and reagents should be directed to and will be fulfilled by the lead contact, Isak S. Pretorius (Sakkie.Pretorius@mq.edu.au) Source data are provided with this paper.

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

## Acknowledgements

The Synthetic Biology initiative led by ISP at Macquarie University is financially supported by the *New South Wales* (NSW) *Government's Department of Primary Industries*, the *Australian Research Council Centre of Excellence in Synthetic Biology*, and external grants from *Bioplatforms Australia*, the *NSW Chief Scientist and Engineer*, and an internal grant from Macquarie University. Ian Paulsen was supported by an *Australian Research Council Laureate Fellowship*. T.C.W. and B.L. were supported by Fellowships from

the *CSIRO Synthetic Biology Future Science Platform* and Macquarie University. T.C.W. and I.S.P. acknowledge the support of *ARC Discovery Project* DP200100717. B.L. acknowledges the support of the *Gordon and Betty Moore Foundation* (GBMF9319, grant), *Twist Bioscience*, and the *Allen Foundation*. Work in the JDB lab was supported by US NSF grants MCB-766 1026068, MCB–1443299, MCB-1616111 and MCB-1921641. Work in the JSB lab was supported by US NSF awards MCB-1445545 and EF-1935355. Some of the research described herein was facilitated by access to the *Australian Proteome Analysis Facility* (APAF) funded under the Australian Government's *National Collaborative Research Infrastructure Strategy* (NCRIS)/ Education Investment Fund. Acknowledgement to Dr Natalie Curach for helping to establish the Sc2.0 project and synthetic biology initiative at Macquarie University, and for helpful discussions, and to Professor Helena Nevalainen for instructive and useful guidance during the project.

## Author contributions

Conceived and coordinated the project (H.D.G., H.K., T.C.W., P.E.E., Y.Z., I.T.P., I.S.P.). Designed *synXVI* (L.A.M., G.S., J.S.B., J.D.B.). Designed experiments (H.D.G., H.K., P.E.E.). Conducted experiments (H.D.G., H.K., P.E., J.C., P.K., J.J.B.T., E.L.I.W., K.P., A.C.C., C.H., S.C., E.L.D., N.V.W., F.H., F.M., R.W., Y.C., R.S.K.W., X.X., M.I.E., B.L.). Analysed data (H.D.G., H.K., P.E.E.). Wrote the manuscript (H.D.G., H.K., P.E.E., I.T.P., I.S.P.). All authors revised and approved the final manuscript.

## Competing interests

T.C.W. and A.C.C. are founders and shareholders of Number 8 Bio Pty Ltd. J.D.B. is a Founder of and consultant to Opentrons LabWorks/ Neochromosome, Inc, and serves or served on the Scientific Advisory Board of the following: CZ Biohub New York, LLC, Logomix, Inc., Modern Meadow, Inc., Rome Therapeutics, Inc., SeaHub, Seattle, WA, Tessera Therapeutics, Inc. and the Wyss Institute. J.S.B. is a Founder of Neochromosome, Inc., and a consultant to Opentrons Labworks, Inc. L.A.M. is a Founder of Neochromosome, Inc., and an employee of Opentrons Labworks, Inc. The remaining authors declare no competing interests.

## Additional information

[1]New South Wales Department of Primary Industries, Elizabeth Macarthur Agriculture Institute, Advanced Gene Technology Centre, Woodbridge Road, Menangle, NSW 2568, Australia. [2]School of Natural Sciences, and ARC Centre of Excellence in Synthetic Biology, Macquarie University, Sydney, Australia. [3]Institute for Systems Genetics, NYU Langone Health, New York, NY 10016, USA. [4]Number 8 Bio, Unit 1A 2/6 Orion Road, Lane Cove West, Sydney, NSW 2066, Australia. [5]The Australian Genome Foundry, Sydney, Australia. [6]Department of Microbiology and Biochemistry, Hochschule Geisenheim University, Geisenheim, Germany. [7]Manchester Institute of Biotechnology, The University of Manchester, 131 Princess Street, Manchester M1 7DN, UK. [8]School of Biological Sciences, The University of Edinburgh, Edinburgh EH9 3BF, UK. [9]Department of Biomedical Engineering, Johns Hopkins University, Baltimore, Maryland 21218, USA. [10]Department of Biochemistry and Molecular Pharmacology, NYU Langone Health, New York, NY 10016, USA. [11]Department of Biomedical Engineering, NYU Tandon School of Engineering, Brooklyn 11201 NY, USA. [12]Present address: Microbiogen Pty. Ltd., Level 4/78 Waterloo Rd, Macquarie Park, Sydney NSW 2113, Australia. [13]These authors contributed equally: Hugh D. Goold, Heinrich Kroukamp. ✉e-mail: hugh.goold@dpi.nsw.gov.au; heinrich.kroukamp@microbiogen.com; ian.paulsen@mq.edu.au; sakkie.pretorius@mq.edu.au

