## [Transparent Peer Review file · Nature Communications]

Construction and iterative redesign of synXVI a 903 kb synthetic *Saccharomyces cerevisiae* chromosome

Corresponding Author: Professor Isak Pretorius

Version 0:

Reviewer comments:

Reviewer #1

(Remarks to the Author)

Goold et al. successfully assembled a 903kb artificial chromosome, replacing the native chromosome XVI in a strain of *S. cerevisiae*. This work is a feat of planning, coordination and construction. They demonstrated negative effects on growth of several specific changes to the chromosomal sequence and improved growth by supplementing these inadvertent deleterious mutations or omissions. The authors conclude with suggestions for the next iteration of design for a synthetic chromosome XVI incorporating the lessons learned about introduced sequences (e.g., *LoxP*sym, PCR tags, stop codon substitutions) interfering with adjacent gene expression and recombination of very large pieces of synthetic DNA.

This work is part of an ongoing effort to create a synthetic version of each chromosome of the model eukaryotic organism, *S. cerevisiae*, and to our knowledge is the first reported successful assembly of a synthetic version of yeast chromosome XVI. The authors propose that a synthetically constructed version of yeast could be useful for industrial fermentation for biomanufacturing and as a tool for deeper understanding of fundamental cell biology. Previous work from other groups is cited as evidence that synthetic genomes might advance biological understanding, but the specific questions likely to be answered by construction of a synthetic yeast genome are not clearly presented. In addition, for bioengineering purposes, strain stability (both genetic and phenotypic) and robustness to growth conditions are critical, but this consideration does not appear to be central to the project design. While SCRaMbLE offers a unique way to randomly rearrange the genome with the potential for increasing target molecule production, the presence of many repeated genomic sequences may lead to unintended, deleterious recombination events. The synthetic chromosome in the paper also clearly reduces the growth rate of the resulting yeast strain, a deficit that remains even after debugging. In general, higher fitness is desirable in an industrial strain, unless a clear engineering advantage arises in tandem with lowered growth rate. Several papers are cited as using SCRaMbLE to increase production titers (references 9-12), but only one of these citations measured final product titer; the others conferred tolerance to stressful environmental conditions, generally resulting from one or two gene deletions. As a result, it's unclear if SCRaMbLE alone provides a clear engineering advantage over traditional methods of library generation such as mutagenesis.

The authors could increase the impact of their work by proposing specific changes to the next version of this synthetic chromosome that address some of these issues. Iterating and building one synthetic chromosome or genome to suit both needs (industrial use and fundamental cell biology) may not be easily done. The Discussion focuses on many of the logistical and technical issues that could be resolved in the next iteration of the synthetic chromosome, but bringing these changes back to the larger purpose of the project would more clearly illustrate their significance; in other words, what is the final goal of going through design-build-test-learn cycles and how would iterating bring you closer to this aim? In addition, how quickly could a new iteration of the synthetic chromosome be constructed, given the increased knowledge and technological advancements since the start of the project?

Generally, the interpretation and conclusions from the data presented are reasonable, with some exceptions. The claim of lower Ctr1 protein abundance (lines 410-416) was not supported by the proteomic data, and the GFP transcription data was not very convincing. Figure 3F shows GFP reporter fluorescence data for strains with different UTR contexts for two genes shown to be key for synthetic strain fitness, CTR1 and GIP3. Although not explicitly stated, it looks like the raw data underlying Figure 3F are presented in Supplementary Figure 3. The flow cytometry data show a bimodal distribution for strains expressing the GFP reporter. This is unexpected – why is the distribution bimodal? Why are so many cells in the population apparently expressing no GFP despite all cells presumably sharing the same reporter construct? These data are

summarized inconsistently as a “mean fluorescence intensity” and as a “percentage of the population expressing GFP”. It’s unclear why one statistic is used in one section of the paper (lines 198-207 of the manuscript), and another statistic is used in another section (lines 224-227). Is the mean fluorescence intensity collected from the entire population for each sample (inappropriate for a bimodal distribution), or just the population that falls within the chosen threshold (more reasonable)? If it is the latter, an SEM contextualizing the variance relative to the effect size is necessary.

Additionally, the fluorescence reporter data for CTR1 do not seem to reconcile with the proteomics data. The fluorescence data support some reduction in CTR1 expression, but the proteomics data say there is no detectable difference between samples. Why is this? The authors use lack of detection of certain genes in the synthetic strain to conclude that the expression of these genes is lower, but what measures were taken to ensure that this is truly because of lower abundance of these specific proteins? The methods imply that total protein was normalized, but further checks that similar amounts of protein were measured for each sample would improve the quality of these data – does a control protein expected to be detected in both samples show up at similar abundances? How does the signal for each individual protein compare to the limit of detection?

For clarity, we suggest a different strain labeling scheme for Figure 3C-F: using the same strain IDs for different strains is confusing (strain i in panels C/D is not the same strain as in panels E/F).

The methodology is mostly appropriate, but we think the spot assays to evaluate fitness could be complemented by a more quantitative approach such as growth rate measurements using flasks or a plate reader. The spot assay shows qualitative major differences in fitness but is not good enough for some of the more subtle fitness differences (especially to support the claims made about tRNA plasmid complementation improving fitness). The colony size difference is slight in this last case, and some quantitative metric for improvement would be valuable. Also, it looks like there are two redundant sections in the Methods describing the spot assay.

In general, there is sufficient detail in the methods for the work to be reproduced, although some places need further clarity:

- Megachunk Preparation: molar ratio of DNA assembly was described on line 663 of manuscript, but it’s not clear what pieces correspond to which part of the ratio (ex. which part is terminal chunk DNA? 4, 2, 1, or 0.5?)
- CRISPR D-BUGS: Line 905 of the manuscript refers to “4 colonies of each” that were analyzed for growth and fitness, but it’s not clear what the each is here – is it 4 colonies per PAM site tested? How many PAM sites were tested total? Looks like data is only shown for 12 strains in Supplementary Figure 2. Is this data for all colonies picked, or just a subset?

As a last general comment, the writing would benefit from editing for clarity and grammar. As a few examples, lines 410-416 and lines 443-452 are difficult to follow. The discussion of tRNA abundance in lines 513-525 is also confusing, using multiple conventions for identifying tRNAs within a few sentences.

Reviewer #2

(Remarks to the Author)

Reviewer #3

(Remarks to the Author)

Despite the substantial effort required by the authors to construct and debug such a large chromosome, I am concerned about the lack of novelty, especially when comparing this paper to other published Sc2.0 papers. The bugs discussed in this paper and the methods used to localize and fix them have already been extensively covered in other SC2.0 publications. If there are novel aspects related to these papers, they need to be better explained in the main text. For example, it would be helpful to highlight how the lessons learned during the synthesis and debugging of synXVI were unique compared to other synthetic chromosomes.

Below are my main criticisms:

Introduction:

The introduction provides many results from the Sc2.0 project and explains how these results enhance our understanding of eukaryotic chromosome complexity and the development of higher strains for biomanufacturing. However, it does not clearly “introduce” how the results of this specific publication will contribute to the scientific community. The abstract claims that “This redesign provides a roadmap into applications of Sc2.0 strategies in non-yeast organisms,” which is appealing to the scientific community beyond Sc2.0. However, the introduction does not clearly explain how this is achievable based on the paper’s results. Although this may be elaborated on later, the connection should begin to be made here.

Page 3, lines 71-72: The statement “Application of synthetic genome design has, to date, been limited largely to bacterial and viral projects” seems unnecessary.

Results:

Page 4, line 112: Is the term "Strain" correct in "Strain A-G (strain 5)"?

Page 5, line 116: "Sequencing read depth revealed multiple assembly errors including tandem chunk and megachunk duplications, deletions, pUC vector insertions, missing TAG-TAA codon swaps, and the presence of wild-type specific introns." Is there data to support this? What code was used for this analysis? It would be helpful to include a supplementary figure showing an example from a segment of the synthetic DNA.

Figure 1: Figure 1 is quite confusing. It would be clearer to present it as two parallel figures—one showing the construction of Strain 9 and the other showing the construction of Strain 12. Then, these two could be combined to create Strain 15. It would also be useful to indicate the position of the centromere.

Page 6, line 143: Was this growth defect observed in strains 7 and 9 (both containing synthetic DNA F3-I4) or strains 11 and 12 (both containing synthetic DNA X3-X4)? It would be great to see these results and compare them with strain 15. Results could be added to Figure 2 or included as a Supplementary Figure.

Page 6, line 160: Where is the data for this comparison? Please reference the data here or include it if it is not present.

Page 7, line 167: Where is the reference for this data? Is there a figure or table? What does "previous backcross data" refer to? Please provide all relevant data and explain how the reader can access it.

Page 7, line 178: Please include data or a schema showing where these crossovers occur. I couldn't find this information.

Page 7, lines 191-192: Refer to the figure with the data here.

Page 8, line 212: "A diploid strain was identified which had similar growth to the parental diploid." Do you have data to support this statement? If so, please provide it. There is no limit to the number of supplementary figures. If it is already present in the manuscript, please reference the figure here.

Page 8, line 216: I couldn't find a strain labeled J7 in Supplementary Figure 4.

Page 8, line 218: I suggest creating a supplementary figure showing the localization of GIP3, dubious ORFs, and adjacent genes, as well as an alignment of sequence data showing the identified crossover. This may help visualize the data discussed here.

Figure 3: This figure is hard to understand due to the use of Roman numerals, which may suggest that they represent the same thing in different plots when they actually do not. Consider replacing them with more direct labels. Also, I recommend labeling strains with unique identifiers rather than referring to them as strain X.

Page 11, line 270: "However, the synthetic chromosome was retained at the same rate as the BY4741 chromosome XVI in a hybrid diploid strain passaged daily over 126 generations (data not shown), demonstrating no instability in chromosomal segregation." Was the chromosome unstable before the changes? This was not discussed or shown, so if you want to keep this statement, it should be compared to previous data. Also, every result discussed in your manuscript should be accompanied by corresponding data. If the data is "not shown," it is better not to include any discussion about it.

Page 12, line 305: If the synthetic MRPL51 gene was not deleted, how do we know that this effect was not caused by MRPL51 over-expression? Would it be better to substitute the synthetic MRPL51 with the wild-type gene and observe the effect? If CTR1 and GIP3 were restored, why was MRPL51 not? Was it?

Page 12, line 309: Why was a whole-cell proteomic approach used instead of a transcriptomic approach, like the one used for GIP3 and CTR1, given that the focus was only on 10 genes?

Page 13, line 320: Do the authors have any insights into why CTR1 and Nvj2 did not show quantitative changes in abundance between BY4741 and the synXVI chromosome of strain 15? This is surprising for CTR1, given that the transcriptomic approach using the GFP reporter showed lower expression in the synthetic version compared to the wild-type. Please include a discussion of this in the manuscript.

Page 14, line 380: Where is Figure 5B?

Discussion:

Page 16, line 413: The author states that "Reduction of GFP activity in a transcriptional reporter assay, and the improvement in the strain's growth with non-fermentable carbon sources in the absence of exogenous CuSO₄ upon restoration of loxPsym sites assigned to AXL1 and YPR123C demonstrates that loxPsym site placement into the 5' UTR results in lower Ctr1 protein abundance." However, the proteomic analysis does not show a difference in Ctr1 protein quantities when comparing strain 15 and BY4741 (wild-type). How should both results (transcriptional reporter assay and proteomic analysis) be interpreted?

Page 16, line 434: Is this result shown in Figure 5 as referenced here?

Page 18, line 478: The authors did not discuss the possibility of using yeast and other organisms to create a library of chromosomes from different species and then transferring them to recipient cells, as demonstrated for mycoplasma (Clyde A. Hutchison III et al. (2006)). Nearly a 1 Mb DNA molecule was assembled in yeast and transferred to human cells (Gambogi et al., (2024) Science). Additionally, techniques that allow for full chromosome deletion/substitution have already been demonstrated for yeast (Coradini et al. (2023) NatComm) and human cells (Bosco et al. (2023) CELL), which may address issues related to sequential chromosome replacement.

Version 1:

Reviewer comments:

Reviewer #1

(Remarks to the Author)

The authors addressed many of the specific technical questions from our review in their revised manuscript. The additional data is welcome, but in at least two cases, the way the new data is presented leaves more questions. For example, Figure 3D now uses a consistent method to measure GFP signal, and triplicate measurements are mentioned but no error bars or standard deviations are shown. I inferred that the measurements shown in the righthand panel of Figure 3D are the medians of the three individual replicates shown as separate points (as a triangle, circle or square), but this is not explained in the caption, nor are any estimates of significance of these results presented. Similarly, growth curves supporting some of the spot assays are now included as Supplementary Figure 9, but no quantitative metrics were added (e.g., growth rates).

Overall, I believe that the data in this manuscript supports the main conclusions described within the paper, and clearly considerable laboratory work underpinned the construction of the synthetic chromosome. As I mentioned in my original review, the manuscript would benefit from additional editing for clarity and grammar. To meet the standard of papers typically published by Nature Communications and maximize impact, I continue to recommend overall editing for concise, clear language, including a compelling explanation of the durable effect of this specific work outside of the Sc2.0 project.

Reviewer #2

(Remarks to the Author)

Reviewer #3

(Remarks to the Author)

I appreciate the authors' effort in addressing all the questions and concerns raised by the reviewers. This effort has significantly improved the manuscript, which now seems ready for publication.

Although not strictly necessary, I believe it would be easier for the audience to discern the phenotypic differences between strains in the added Supplementary Figure 9 if a more direct growth parameter, such as doubling time or maximum growth rate (u_{max}), were shown instead of the raw growth curve."

**Reviewer 1**

**Comment 1.1:**

Goold et al. successfully assembled a 903kb artificial chromosome, replacing the native chromosome
XVI in a strain of *S. cerevisiae*. This work is a feat of planning, coordination and construction. They
demonstrated negative effects on growth of several specific changes to the chromosomal sequence
and improved growth by supplementing these inadvertent deleterious mutations or omissions. The
authors conclude with suggestions for the next iteration of design for a synthetic chromosome XVI
incorporating the lessons learned about introduced sequences (e.g., LoxPsym, PCR tags, stop codon
substitutions) interfering with adjacent gene expression and recombination of very large pieces of
synthetic DNA.

This work is part of an ongoing effort to create a synthetic version of each chromosome of the model
eukaryotic organism, *S. cerevisiae*, and to our knowledge is the first reported successful assembly of a
synthetic version of yeast chromosome XVI. The authors propose that a synthetically constructed
version of yeast could be useful for industrial fermentation for biomanufacturing and as a tool for
deeper understanding of fundamental cell biology. Previous work from other groups is cited as
evidence that synthetic genomes might advance biological understanding, but the specific questions
likely to be answered by construction of a synthetic yeast genome are not clearly presented. In
addition, for bioengineering purposes, strain stability (both genetic and phenotypic) and robustness to
growth conditions are critical, but this consideration does not appear to be central to the project
design. While SCRaMbLE offers a unique way to randomly rearrange the genome with the potential
for increasing target molecule production, the presence of many repeated genomic sequences may
lead to unintended, deleterious recombination events. The synthetic chromosome in the paper also
clearly reduces the growth rate of the resulting yeast strain, a deficit that remains even after
debugging. In general, higher fitness is desirable in an industrial strain, unless a clear engineering
advantage arises in tandem with lowered growth rate. Several papers are cited as using SCRaMbLE to
increase production titers (references 9-12), but only one of these citations measured final product
titer; the others conferred tolerance to stressful environmental conditions, generally resulting from one
or two gene deletions. As a result, it's unclear if SCRaMbLE alone provides a clear engineering
advantage over traditional methods of library generation such as mutagenesis.

**Response 1.1:**

We've changed the text to better address the use of Sc2.0 for strain improvement, more broadly than
just titres (Line 65-66 "pathway optimization can be seen in several examples where SCRaMbLE has
been applied to increase titre of a final product yield or robustness of strains"). Also, the reviewer
raised a point we have not discussed but should include. The benefit of taking a SCRaMbLE approach
over a classic EMS or UV mutagenesis approach is that it allows the exploration of a different
evolutionary landscape. EMS or UV mutagenesis typically result in SNPs or indel mutations, while
SCRaMbLE enables large scale events, such as deletions, duplications, inversions and
rearrangements, which are normally rare events that typically happen only over evolutionary
timescales. Thus, EMS or UV mutagenesis are well suited for selecting for simple phenotypes, such
as increasing the expression of a single gene or tweaking the kinetics of a single enzyme. SCRaMbLE
opens the possibility of evolving more complex phenotypic changes that require changes in
expression or function of multiple genes. We have expanded the discussion to mention use of more
robust strains in future to improve industrial relevance of the Sc2.0 project.

**Comment 1.2:**

The authors could increase the impact of their work by proposing specific changes to the next version
of this synthetic chromosome that address some of these issues. Iterating and building one synthetic
chromosome or genome to suit both needs (industrial use and fundamental cell biology) may not be
easily done. The Discussion focuses on many of the logistical and technical issues that could be
resolved in the next iteration of the synthetic chromosome, but bringing these changes back to the
larger purpose of the project would more clearly illustrate their significance; in other words, what is
the final goal of going through design-build-test-learn cycles and how would iterating bring you
closer to this aim? In addition, how quickly could a new iteration of the synthetic chromosome be
constructed, given the increased knowledge and technological advancements since the start of the
project?

**Response 1.2:**

This is a great point, the major challenge with synXVI as well as some of the other chromosomes was
that with the large size (approaching 1Mb). Even with powerful debugging tools like CRISPR D-
BUGS large chromosomal constructs can be difficult to build and to debug. If there are two loci with
deleterious phenotypes, restoration of one locus may not be sufficient to improve cell fitness. Taken
together with the insertion of markers in coding sequences, promoters or other sensitive loci, one
valuable suggestion mentioned in a recent article about synthetic genomics, could be to design
genomes with more smaller chromosomes. Dividing a new synthetic yeast genome into 60 200Mb
chromosomes, for example, may be easier for human designs and debugging. This may create new
problems but is an objective worth discussing. It is difficult to predict how quickly a new

chromosome construction could take given the unpredictable nature of biology, however it would be
unlikely to be longer than 1 month per megachunk, which can be built in tandem for longer
chromosomes, so it is dependent on the size of the synthetic chromosome. We have elaborated in the
discussion on other projects such as CReATiNG which have been used similarly to more expediently
build large chromosomes.

**Comment 1.3:**

Generally, the interpretation and conclusions from the data presented are reasonable, with some
exceptions. The claim of lower Ctr1 protein abundance (lines 410-416) was not supported by the
proteomic data, and the GFP transcription data was not very convincing. Figure 3F shows GFP
reporter fluorescence data for strains with different UTR contexts for two genes shown to be key for
synthetic strain fitness, CTR1 and GIP3. Although not explicitly stated, it looks like the raw data
underlying Figure 3F are presented in Supplementary Figure 3. The flow cytometry data show a
bimodal distribution for strains expressing the GFP reporter. This is unexpected – why is the
distribution bimodal? Why are so many cells in the population apparently expressing no GFP despite
all cells presumably sharing the same reporter construct? These data are summarized inconsistently as
a “mean fluorescence intensity” and as a “percentage of the population expressing GFP”. It’s unclear
why one statistic is used in one section of the paper (lines 198-207 of the manuscript), and another
statistic is used in another section (lines 224-227). Is the mean fluorescence intensity collected from
the entire population for each sample (inappropriate for a bimodal distribution), or just the population
that falls within the chosen threshold (more reasonable)? If it is the latter, an SEM contextualizing the
variance relative to the effect size is necessary.

**Response 1.3:**

The data have been updated to represent the mean fluorescence intensity of the GFP⁺ gated cells
between the wildtype PRS mock populations, and the various different GFP expressing cells, and the
text references this change more clearly.

The text has been updated to more clearly explain that the phenotypes are results of interruption of
dynamic regulation and not necessarily due to downregulation. Specifically, *CTR1* is required for cell
survival on non-fermentable carbon sources. In conditions of high levels of copper, during
fermentative growth it is heavily downregulated. Under conditions of respiratory growth and also
during DNA damage, however, copper is required for function of cytochrome C oxidase in the
mitochondrial electron transport chain, active during respiratory growth and thus essential for growth
on non-fermentable carbon sources. The interrupted transcription start site is likely inhibiting the
induction of *CTR1* expression during growth on non-fermentable carbon sources leading to cell death.

Examples shown in Figure 3 and in reference Wu, X., Sinani, D., Kim, H. & Lee, J. *Copper Transport*
*Activity of Yeast Ctr1 Is Down-regulated via Its C Terminus in Response to Excess Copper. Journal of*
*Biological Chemistry* 284, 4112–4122 (2009) demonstrate the inability of *CTR1*Δ strains to grow on
non-fermentable carbon sources without supplementation of exogenous copper.

Bimodal distribution in yeast cells expressing *GFP* is common and is dependent on the specific
promoter, fluorescent protein, or the cellular conditions [for example see 1. *Yeast Protocols. vol. 1163*
*(Springer New York, New York, NY, 2014)*. 2. Williams, T. C., Xu, X., Ostrowski, M., Pretorius, I. S. &
Paulsen, I. T. *Positive-feedback, ratiometric biosensor expression improves high-throughput*
*metabolite-producer screening efficiency in yeast. Synthetic Biology* 2, ysw002 (2017)]. There are
various explanations for this phenomenon, pRS vectors bearing centromeric origins of replication are
known to have copy numbers between 1 and 3, which may vary within a population, cells in a
population may be at different stages of the cell cycle and dynamic expression of *CTR1p* may vary
across those populations etc.

Very low *GFP* fluorescence driven by highly repressed promoters would likely fall below the
detection threshold of the flow cytometer and hence look like "cells in the population apparently
expressing no *GFP*".

We have updated the manuscript text to consistently use “median fluorescence intensity” referring to
cells falling within the *GFP*-positive gate.

**Comment 1.4:**

Additionally, the fluorescence reporter data for *CTR1* do not seem to reconcile with the proteomics
data. The fluorescence data support some reduction in *CTR1* expression, but the proteomics data say
there is no detectible difference between samples. Why is this? The authors use lack of detection of
certain genes in the synthetic strain to conclude that the expression of these genes is lower, but what
measures were taken to ensure that this is truly because of lower abundance of these specific proteins?
The methods imply that total protein was normalized, but further checks that similar amounts of
protein were measured for each sample would improve the quality of these data – does a control
protein expected to be detected in both samples show up at similar abundances? How does the signal
for each individual protein compare to the limit of detection?

**Response 1.4:**

Cellular copper transport in yeast is under careful regulatory control, as copper is cytotoxic, but is also
required post-diauxic shift when the cell needs to rely on the mitochondrial cytochrome C oxidase

(CcO) for respiratory growth on ethanol. Thus, expression levels of *CTR1* are likely to be low under
initial growth conditions, but will be induced post-diauxic shift, or when grown on glycerol as a sole
carbon source (Wu, X., Sinani, D., Kim, H. & Lee, J. Copper Transport Activity of Yeast Ctr1 Is
Down-regulated via Its C Terminus in Response to Excess Copper. *Journal of Biological Chemistry*
284, 4112–4122 (2009).

In Strains 29 and 30, the transcription start site of *CTR1* is interrupted by a loxPsym site, which likely
impacts either expression or regulation of *CTR1*. In panel D of Figure 3 we have altered the
representation of the GFP-transcriptional reporter assay to include the individual data points to make
it clearer that the aberrant expression of *CTR1* in strains with the loxPsym site associated with
YPR123C. This is consistent with our observation (Figure 3, Panel C) that the cells with a loxP site
adjacent to the *CTR1* transcription start site display a phenotype identical to a *CTR1* knockout, which
can also be rescued with exogenous copper.

Reviewer 1 has pointed out that the non-detection of proteins, particularly those of low abundance,
may not represent significant changes to protein abundance. The global proteomic analysis was an
effort to determine whether or not there was a global pattern related to protein abundance. We have
reviewed the proteomic data in light of the Reviewer's concerns, and we note that three out of the four
proteins that were detected in the BY4741 parental strain, but not in Strain 15, were in relatively low
abundance in BY4741. We agree with the Reviewer that this opens up the possibility that we can't be
certain that this is a significant difference in expression between the two strains. Given this issue, we
have greatly reduced this text associated with the proteomic work. The role of these genes in synXVI
fitness is subsequently addressed by complementation assays later in the text.

With regards to the query about control proteins in the proteomics data, we checked the abundances of
classically 'constitutively expressed' proteins Pgc1 and Tef1, which showed similar abundances in
both samples, with no statistically significant differences.

Broadly, our flow cytometry data demonstrate that disruption of the Transcriptional Start Site (TSS)
impacts regulatory control of the genes, and our experimental growth data demonstrates that restoring
these sites on the chromosome improves growth. Chantal Yue Shen and Jef D Boeke have
communicated that due to their palindromic sequence loxPsym sites flanking genes create aberrant
transcriptional units where RNA molecules can create hairpin loop structures which interfere with
proper translation and that this is unpublished work. It is possible that the second flanking loxPsym
sites associated with *CTR1* and *GIP3* on the chromosome exacerbates disruption of transcriptional
regulation or of translation.

**Comment 1.5:**

For clarity, we suggest a different strain labeling scheme for Figure 3C-F: using the same strain IDs
for different strains is confusing (strain i in panels C/D is not the same strain as in panels E/F).

**Response 1.5:**

**Figure 3 has been redesigned to account for suggestions by both reviewers.**

**Comment 1.6:**

The methodology is mostly appropriate, but we think the spot assays to evaluate fitness could be
complemented by a more quantitative approach such as growth rate measurements using flasks or a
plate reader. The spot assay shows qualitative major differences in fitness but is not good enough for
some of the more subtle fitness differences (especially to support the claims made about tRNA
plasmid complementation improving fitness). The colony size difference is slight in this last case, and
some quantitative metric for improvement would be valuable. Also, it looks like there are two
redundant sections in the Methods describing the spot assay.

**Response 1.6:**

**We agree with the reviewer's comments and have included growth curves to demonstrate gradual
improvement through our iterations of DBTL with synXVI in the supplementary data (Supplementary
Figure 9). Spot assays have been used throughout the Sc2.0 project and we are obliged to follow
certain protocols such as spot assays to maintain consistency across the consortium.**

**Comment 1.7:**

In general, there is sufficient detail in the methods for the work to be reproduced, although some
places need further clarity:

**Comment 1.7.1:** Megachunk Preparation: molar ratio of DNA assembly was described on line 663 of
manuscript, but it's not clear what pieces correspond to which part of the ratio (ex. which part is
terminal chunk DNA? 4, 2, 1, or 0.5?)

**Response 1.71:** This has been clarified in text.

**Comment 1.7.2:** CRISPR D-BUGS: Line 905 of the manuscript refers to "4 colonies of each" that
were analyzed for growth and fitness, but it's not clear what the each is here – is it 4 colonies per
PAM site tested? How many PAM sites were tested total? Looks like data is only shown for 12 strains
in Supplementary Figure 2. Is this data for all colonies picked, or just a subset?

**Response 1.7.2:** An initial screen of (ideally) four colonies per PAM site tested were conducted. If
these colonies had varied phenotypes, further colonies were screened to improve the probability of
identification of the causative genetic locus. We have modified the manuscript text to clarify this. We
have not included every spot assay conducted as the images number in the thousands and do not add
worthwhile information to the manuscript.

**Comment 1.8:**

As a last general comment, the writing would benefit from editing for clarity and grammar. As a few
examples, lines 410-416 and lines 443-452 are difficult to follow. The discussion of tRNA abundance
in lines 513-525 is also confusing, using multiple conventions for identifying tRNAs within a few
sentences.

**Response 1.8**

Language has been simplified, and tRNA naming convention has been normalised across the article.

**Reviewer 2:**

**Comment 2.1:**

I co-reviewed this manuscript with one of the reviewers who provided the listed reports. This is part
of the *Nature Communications* initiative to facilitate training in peer review and to provide
appropriate recognition for Early Career Researchers who co-review manuscripts.

**Response 2.1:**

We thank Reviewer 2 and the co-reviewer for assessing this manuscript.

**Reviewer 3:**

**Comment 3.1:**

Despite the substantial effort required by the authors to construct and debug such a large
chromosome, I am concerned about the lack of novelty, especially when comparing this paper to other
published Sc2.0 papers. The bugs discussed in this paper and the methods used to localize and fix
them have already been extensively covered in other SC2.0 publications. If there are novel aspects
related to these papers, they need to be better explained in the main text. For example, it would be

helpful to highlight how the lessons learned during the synthesis and debugging of synXVI were
unique compared to other synthetic chromosomes.

As one of the largest and last chromosomes to be completed in Sc2.0, this has given us a unique
opportunity to reflect on the lessons learnt from chromosome construction and genome design. The
lessons learnt from our SC2.0 experience are presented in the section “redesign of SynXVI and in
Figure 5”. The lessons from SynXVI and synthesized from other synthetic chromosomes have not
been discussed in earlier Sc2.0 projects and we believe the redesign we present is a novel and
important iteration of the first eukaryotic genome synthesized. In light of recent literature looking to
design eukaryotic genomes, this is an indispensable resource for guiding future genome or
chromosome design.

Below are my main criticisms:

*Introduction:*

The introduction provides many results from the Sc2.0 project and explains how these results enhance
our understanding of eukaryotic chromosome complexity and the development of higher strains for
biomanufacturing. However, it does not clearly "introduce" how the results of this specific publication
will contribute to the scientific community. The abstract claims that “This redesign provides a
roadmap into applications of Sc2.0 strategies in non-yeast organisms,” which is appealing to the
scientific community beyond Sc2.0. However, the introduction does not clearly explain how this is
achievable based on the paper’s results. Although this may be elaborated on later, the connection
should begin to be made here.

**Response 3.1:**

We thank Reviewer 3 for their constructive comments.

We have included a summary statement in the *Introduction* “Given growing interest in synthetic
genomics with the commencement of projects with plant genomes and other organisms, we outline
lessons learnt from SynXVI and Yeast 2.0 more broadly, and present a refined version of synXVI as
an example of iterated genome design parameters.” and we also highlight the relatively rare problem
in the Sc2.0 project of building chromosomes of particularly large size (1 Mb) which have higher co-
incidence of defective loci simply due to the nature of their size compared to the smaller
chromosomes (e.g., Chr I, III, and VI which are all smaller than 320 kb). Specifically, we mention in
the discussion the idea of redesigning genomes to have more smaller chromosomes rather than fewer
large chromosomes, and the benefits of that approach despite the challenge of potentially introducing
aneuploidy or issues with insufficient numbers of centromeres.

**Comment 3.2:**

Page 3, lines 71-72: The statement “Application of synthetic genome design has, to date, been limited
largely to bacterial and viral projects” seems unnecessary.

**Response 3.2:**

We feel it is valuable to highlight that this is one part of a larger and world-first project of complete
construction of a whole *eukaryotic* genome design. We prefer to retain this statement.

**Comment 3.3:**

*Results:*

Page 4, line 112: Is the term "Strain" correct in “Strain A-G (strain 5)”?

**Response 3.3:**

This section has been rewritten to characterise by strain number with the synthetic megachunks
referred to in parentheses.

**Comment 3.4:**

Page 5, line 116: “Sequencing read depth revealed multiple assembly errors including tandem chunk
and megachunk duplications, deletions, pUC vector insertions, missing TAG-TAA codon swaps, and
the presence of wild-type specific introns.” Is there data to support this? What code was used for this
analysis? It would be helpful to include a supplementary figure showing an example from a segment
of the synthetic DNA.

**Response 3.4:**

We have included an example supplementary figure (Supplementary Figure 1) showing the use of
Geneious mapper to use read overage to identify a duplication and deletion in one of the early strains,
the raw read data will be available on publication through NCBI.

**Comment 3.5:**

*Figure 1:* Figure 1 is quite confusing. It would be clearer to present it as two parallel figures—one
showing the construction of Strain 9 and the other showing the construction of Strain 12. Then, these
two could be combined to create Strain 15. It would also be useful to indicate the position of the
centromere.

**Response 3.5:**

Figure 1 has been redesigned for clarity.

Comment 3.6:

Page 6, line 143: Was this growth defect observed in strains 7 and 9 (both containing synthetic DNA F3-I4) or strains 11 and 12 (both containing synthetic DNA X3-X4)? It would be great to see these results and compare them with strain 15. Results could be added to Figure 2 or included as a Supplementary Figure.

Response 3.6:

As detailed in the discussion a problem with the construction of *synXVI* was the multiple genes causing deleterious growth, including the generation of ρ^- mutants. Strains 7 and 9 are difficult to compare analyse for these purposes as they have very poor growth. Similarly, a point mutation in *RRP12* on strains 10 and 12 means this strain also exhibited very poor growth on glycerol and at 37°C. The inclusion of the synthetic *CTR1* sequence in a BY4741 background in Figure 3, Panel C clearly demonstrates the deleterious growth associated with that locus.

Comment 3.7:

Page 6, line 160: Where is the data for this comparison? Please reference the data here or include it if it is not present.

Response 3.7:

These data are located in supplementary Figures 2 and 4, and the manuscript has been updated for clarity.

Comment 3.8:

Page 7, line 167: Where is the reference for this data? Is there a Figure or Table? What does “previous backcross data” refer to? Please provide all relevant data and explain how the reader can access it.

Response 3.8:

The text now refers to Figure 3 Panels A and B.

Comment 3.9:

Page 7, line 178: Please include data or a schema showing where these crossovers occur. I couldn't find this information.

**Response 3.9:**

A schematic diagram has been included in Supplementary data to depict the crossovers in the healthy
grower strains from this CRISPR D-BUGS (Supplementary Figure 2).

**Comment 3.10:**

*Page 7, lines 191-192:* Refer to the Figure with the data here.

**Response 3.10:**

Figure reference has been included in the text.

**Comment 3.11:**

Page 8, line 212: “A diploid strain was identified which had similar growth to the parental diploid.”

Do you have data to support this statement? If so, please provide it. There is no limit to the number of
supplementary figures. If it is already present in the manuscript, please reference the figure here.

**Response 3.11:**

This was poorly worded and we have rectified it. Of the seven single colonies identified from the
CRISPR D-BUGS screen, one had much better growth than the parental heterodiploid strain 27, this is
colony 7, with fitness approaching the BY4741 control used to demonstrate the ideal growth
phenotype. The sequencing revealed a localised chromosomal crossover removing a loxPsym site
from the synthetic chromosome, yielding a homozygous wildtype locus adjacent to *GIP3*.

**Comment 3.12:**

*Page 8, line 216:* I couldn't find a strain labelled J7 in Supplementary Figure 4.

**Response 3.12:**

The Figure has been updated to Colony 7, and so the text has now been updated to appropriately refer
to Colony 7.

**Comment 3.13:**

*Page 8, line 218:* I suggest creating a supplementary figure showing the localization of GIP3, dubious
ORFs, and adjacent genes, as well as an alignment of sequence data showing the identified crossover.

This may help visualize the data discussed here.

**Response 3.13:**

This has been included in a new supplementary figure along with schematic diagrams depicting
*AXLI/CTR1* (Supplementary Figure 6)

**Comment 3.14:**

*Figure 3:* This Figure is hard to understand due to the use of Roman numerals, which may suggest
that they represent the same thing in different plots when they actually do not. Consider replacing
them with more direct labels. Also, I recommend labeling strains with unique identifiers rather than
referring to them as strain X.

**Response 3.14:**

Figure 3 has been redesigned to account for suggestions by both reviewers.

**Comment 3.15:**

*Page 11, line 270:* “However, the synthetic chromosome was retained at the same rate as the BY4741
chromosome XVI in a hybrid diploid strain passaged daily over 126 generations (data not shown),
demonstrating no instability in chromosomal segregation.” Was the chromosome unstable before the
changes? This was not discussed or shown, so if you want to keep this statement, it should be
compared to previous data. Also, every result discussed in your manuscript should be accompanied by
corresponding data. If the data is “not shown,” it is better not to include any discussion about it.

**Response 3.15:**

This reference has been removed.

**Comment 3.16:**

*Page 12, line 305:* If the synthetic *MRPL51* gene was not deleted, how do we know that this effect
was not caused by *MRPL51* over-expression? Would it be better to substitute the synthetic *MRPL51*
with the wild-type gene and observe the effect? If *CTR1* and *GIP3* were restored, why was *MRPL51*
not? Was it?

**Response 3.16:**

Since submission of the original manuscript which included data of complementation of *MRPL51*
expressed under native control on a pRS413 vector, wildtype *MRPL51* has been reintroduced into the
chromosome, and growth curves (Supplementary Figure 9) have been used to demonstrate improved
fitness between *MRPL51* synthetic (strain 20) and *MRPL51* wildtype (strain 33).

**Comment 3.17:**

*Page 12, line 309:* Why was a whole-cell proteomic approach used instead of a transcriptomic
approach, like the one used for GIP3 and CTR1, given that the focus was only on 10 genes?

**Response 3.17:**

*CTR1* was initially identified through backcross data and CRISPR D-BUGS as a defect-causing locus,
and *GIP3* was identified similarly through CRISPR D-BUGS. After a pattern of defects was identified
we pursued single cell proteomics to identify whether or not similar patterns existed across the
genome in other proteins with similar loxPsym site placements. While, as we see with *CTR1*, protein
abundance may not be significantly different in normal growth conditions, we supposed there may
have been significant differences in other proteins from genes encoded across chromosome XVI. We
used a global proteomics approach to also see if there were any other unexpected cellular impacts
resulting from *synXVI*.

**Comment 3.18:**

*Page 13, line 320:* Do the authors have any insights into why CTR1 and Nvj2 did not show
quantitative changes in abundance between BY4741 and the *synXVI* chromosome of strain 15? This
is surprising for CTR1, given that the transcriptomic approach using the GFP reporter showed lower
expression in the synthetic version compared to the wild-type. Please include a discussion of this in
the manuscript.

**Response 3.18:**

As discussed in our responses addressing other points, we believe the dynamic control of the
expression of genes flanked by loxPsym sites is altered. In the case of *CTR1* this is corroborated by
the phenotype of Strain 15 and the *CTR1* knockout strain referred to in our rebuttal of a similar point
made by Reviewer 1, referencing Figure 3.

**Comment 3.19:**

*Page 14, line 380:* Where is Figure 5B?

**Response 3.19:**

Figure 5 was originally set out in two panels, A and B, this has been rectified to match the existing
Figure 5.

**Comment 3.20:**

*Discussion:*

*Page 16, line 413:* The author states that “Reduction of GFP activity in a transcriptional reporter
assay, and the improvement in the strain’s growth with non-fermentable carbon sources in the absence
of exogenous CuSO₄ upon restoration of loxPsym sites assigned to AXL1 and YPR123C
demonstrates that loxPsym site placement into the 5’ UTR results in lower Ctr1 protein abundance.”
However, the proteomic analysis does not show a difference in Ctr1 protein quantities when
comparing strain 15 and BY4741 (wild-type). How should both results (transcriptional reporter assay
and proteomic analysis) be interpreted?

**Response 3.20:**

We agree with the reviewer that the phrasing is inexact and have changed it to read "... loxPsym site
placement into the 5’ UTR impacts *CTR1* expression or regulation". With regards to the differences in
transcriptional reporter assay and proteomic analysis, please see our detailed response to Reviewer 1
regarding Ctr1.

We believe that *CTR1* induction is altered. The growth phenotype (Figure 3) suggests that the
gene is not being induced properly on growth on non-fermentable carbon sources. This is consistent
with data from other publications (cited above) which show recovery of growth on non-fermentable
carbon sources with the addition of exogenous copper. Copper is toxic to the cell and is not essential
to fermentative growth, however for function of mitochondrial *CcO* during respiratory growth
conditions copper is essential. Given the *synCTR1* cells in figure 3 behave identically to reported
*CTR1Δ* on glucose, glycerol, and glycerol supplemented with copper, we believe induction of *CTR1* is
interrupted during growth on non-fermentable carbon sources. This is experimentally difficult to
prove as these cells are inviable, and in normal growth conditions *CTR1* expression is repressed.

We invite reviewer 3 to refer to our rebuttal to reviewer one, briefly, we have demonstrated
aberrant dynamic control of *CTR1* in Figure 3, where absence of growth of strains bearing synthetic
*CTR1* matches a *CTR1* knockout discussed in literature which is unable to mobilise copper when it is
required for mitochondrial function during growth on non-fermentable carbon sources.

**Comment 3.21:**

Page 16, line 434: Is this result shown in Figure 5 as referenced here?

**Response 3.21:**

Thank you to the reviewer for pointing this out, the passage erroneously referenced Figure 5, not
Supplementary Figure 7 and so we have updated the Figure reference.

**Comment 3.22:**

*Page 18, line 478:* The authors did not discuss the possibility of using yeast and other organisms to
create a library of chromosomes from different species and then transferring them to recipient cells, as
demonstrated for mycoplasma (Clyde A. Hutchison III et al. (2006)). Nearly a 1 Mb DNA molecule
was assembled in yeast and transferred to human cells (Gambogi et al., (2024) Science). Additionally,
techniques that allow for full chromosome deletion/substitution have already been demonstrated for
yeast (Coradini et al. (2023) NatComm) and human cells (Bosco et al. (2023) CELL), which may
address issues related to sequential chromosome replacement.

**Response 3.22:**

This is worth discussing; however, it is a complicated proposal which we feel is beyond the scope of
this communication. In particular with heavily regulated mitochondrial proteins where
overexpression may cause phenotypic issues (just as with deletion or under expression). Working with
foreign DNA from plants, algae, and mammals where the genes pose a lower risk of physiological
interruption of the host cells, is a fantastic way of building chromosomes, however when building
yeast chromosomes in yeast cells the challenges around deleterious physiology will always pose a
risk. We have incorporated the references suggested by the reviewer, and thank them for their
suggestions.

RESPONSE TO REVIEWERS' FINAL COMMENTS

Reviewer #1 (Remarks to the Author):

The authors addressed many of the specific technical questions from our review in their revised manuscript. The additional data is welcome, but in at least two cases, the way the new data is presented leaves more questions. For example, Figure 3D now uses a consistent method to measure GFP signal, and triplicate measurements are mentioned but no error bars or standard deviations are shown. I inferred that the measurements shown in the righthand panel of Figure 3D are the medians of the three individual replicates shown as separate points (as a triangle, circle or square), but this is not explained in the caption, nor are any estimates of significance of these results presented. Similarly, growth curves supporting some of the spot assays are now included as Supplementary Figure 9, but no quantitative metrics were added (e.g., growth rates).

Overall, I believe that the data in this manuscript supports the main conclusions described within the paper, and clearly considerable laboratory work underpinned the construction of the synthetic chromosome. As I mentioned in my original review, the manuscript would benefit from additional editing for clarity and grammar. To meet the standard of papers typically published by Nature Communications and maximize impact, I continue to recommend overall editing for concise, clear language, including a compelling explanation of the durable effect of this specific work outside of the Sc2.0 project.

Reviewer #2 (Remarks to the Author):

Reviewer #3 (Remarks to the Author):

I appreciate the authors' effort in addressing all the questions and concerns raised by the reviewers. This effort has significantly improved the manuscript, which now seems ready for publication.

Although not strictly necessary, I believe it would be easier for the audience to discern the phenotypic differences between strains in the added Supplementary Figure 9 if a more direct growth parameter, such as doubling time or maximum growth rate (μ_{max}), were shown instead of the raw growth curve."

RESPONSE: Thank you to the reviewers for their helpful critique which has greatly improved the quality of the article. In Figure 3 the points, a triangle, circle and square, which represent the median fluorescence intensity of the population within GFP⁺ gated cells from three biological replicate cultures, we have included the source data and the flow plots in the supplementary data section of this paper which is more meaningful than standard deviation for flow cytometry datasets. We have also updated supplementary Figure 9 to include μ_{Max} .